# Bayesian neural networks for stock price forecasting before and during COVID-19 pandemic

Rohitash Chandra◉✉*, Yixuan He◉

School of Mathematics and Statistics, University of New South Wales, Sydney, Australia

◉ These authors contributed equally to this work.
* rohitash.chandra@unsw.edu.au

**Data Availability Statement:** We note that data and code used in the paper is open and processed data has been provided via Github repo with link given in the paper: https://github.com/sydney-machine-learning/Bayesianneuralnet_stockmarket.

## Abstract

Recently, there has been much attention in the use of machine learning methods, particularly deep learning for stock price prediction. A major limitation of conventional deep learning is uncertainty quantification in predictions which affect investor confidence. Bayesian neural networks feature Bayesian inference for providing inference (training) of model parameters that provides a rigorous methodology for uncertainty quantification in predictions. Markov Chain Monte Carlo (MCMC) sampling methods have been prominent in implementing inference of Bayesian neural networks; however certain limitations existed due to a large number of parameters and the need for better computational resources. Recently, there has been much progress in the area of Bayesian neural networks given the use of Langevin gradients with parallel tempering MCMC that can be implemented in a parallel computing environment. The COVID-19 pandemic had a drastic impact in the world economy and stock markets given different levels of lockdowns due to rise and fall of daily infections. It is important to investigate the performance of related forecasting models during the COVID-19 pandemic given the volatility in stock markets. In this paper, we use novel Bayesian neural networks for multi-step-ahead stock price forecasting before and during COVID-19. We also investigate if the pre-COVID-19 datasets are useful of modelling stock price forecasting during COVID-19. Our results indicate due to high volatility in the stock-price during COVID-19, it is more challenging to provide forecasting. However, we found that Bayesian neural networks could provide reasonable predictions with uncertainty quantification despite high market volatility during the first peak of the COVID-19 pandemic.

## 1 Introduction

Stock price prediction is a challenging research area [1] due to multiple factors affecting the stock market that range from politics [2], weather and climate, and international and regional trade [3]. Machine learning methods such as neural networks have been widely used in stock forecasting [4]. Some studies show that neural networks outperforms statistical methods, such as multiple linear regression analysis [5], discriminant analysis [6] and related methods.

**Funding:** The authors received no specific funding for this work.

**Competing interests:** The authors have declared that no competing interests exist.

Obtaining robust quantification of uncertainty with good prediction accuracy has been a major challenge for effective stock market prediction models.

Bayesian inference offers a methodology for robust estimation uncertainty quantification of parameters in prediction models. Bayesian inference enables models to feature uncertainty quantification in predictions using posterior distributions to represent the unknown parameters [7]. The probability of hypothesis in Bayesian inference is updated with evidence or data as it become available [8]. We can obtain the posterior distribution by sampling methods that take into account the prior distribution and the likelihood function to evaluate the model with given data [9]. Markov Chain Monte Carlo (MCMC) methods implement Bayesian inference by sampling from the posterior distribution [7, 10]. Bayesian neural networks use MCMC methods to estimate (train) neural network parameters (weights and biases) [11, 12]. MCMC methods face limitations as the size of model and data increase, due to the *curse of dimensionality*. Therefore, Hamiltonian MCMC [13] and Langevin dynamics based MCMC [14] that feature gradient-based proposal distributions were proposed to improve canonical MCMC methods. Parallel tempering MCMC with Langevin-gradients has been very promising for Bayesian neural networks [15]; therefore, it has the potential for forecasting the stock market.

The *coronavirus disease 2019* (COVID-19) is an infectious disease [16–18] which became a global pandemic [19]. The first confirmed or index case of COVID-19 was traced back to 17th November in Wuhan, Hubei, China that became known in December 2019 [20]. The COVID-19 pandemic forced many countries to close their borders and enforce a partial or full lock down which had a devastating impact on the world economy that can continue for years to follow [21–23]. The lock downs and economic impact affected population depending on agriculture in rural areas, especially in developing countries [24, 25]. Several machine learning methods have been utilized for COVID-19 infection prediction in several countries [26–33], and the impact of COVID-19 on the economy has also been studied.

In the literature, there is no work done using Bayesian neural networks for stock markets to provide uncertainty quantification in predictions. We can leverage these methods to harness power of neural networks in prediction accuracy and also to quantify uncertainty in predictions. Moreover, it is worthwhile to evaluate how neural network models perform during the COVID-19 pandemic given drastic changes in the international stock market with disruptions in international trade and prediction. Hence, it is important to investigate the performance of related forecasting models during the COVID-19 pandemic given volatility in stock markets.

In this paper, we use novel Bayesian neural networks for multi-step-ahead stock price prediction before and during COVID-19. We compare our forecasting results with state-of-art neural network training algorithms. Our training data features stocks from four different regions, that include Germany, China, Australia and the United States. We select stocks from these countries due to the effect of various types of lock downs during the course of the first phase of the COVID-19 pandemic that affected their gross domestic product (GDP). We restrict our study to selected stocks from these countries due to geopolitical status and diverse GDP forecasts during the pandemic [23]. Moreover, one of the selected stocks feature COVID-19 mask manufacturing company, while others feature regarding luxury goods for a diverse stock-market analysis. We investigate if the pre-COVID-19 datasets can provide any insights in modelling stock price forecasting during first phase of COVID-19. We compare the prediction performance pre-COVID-19 with results during COVID-19 to evaluate the ability of Bayesian neural networks given drastic changes in the stock price.

We note that although there are many studies in the literature regarding COVID-19 forecasting with machine learning methods, the use of Bayesian neural networks is limited. Moreover, most studies focus on the spread of the infections rather than stock market prediction.

The novelty in this study lies in the use of Bayesian neural networks which provides better model uncertainty quantification when compared to classical neural networks.

The rest of the paper is organised as follows. Section 2 presents a review of related work. Section 3 presents the proposed methodology and Section 4 presents experiments and results. Section 5 provides a discussion and Section 6 concludes the paper with discussion of future work.

## 2 Related work

### 2.1 Stock market and price forecasting

There are two types of forecasting methods, which are fundamental analysis techniques [34] and technical analysis [35]. Fundamental analysis techniques measure intrinsic value of the stock by examining relevant data of the company such as audit reports, book value and price-to-earnings ratio (P/E ratio) [36]. Technical analysis attempts to predict stock markets using charts and quantitative indicators [37]. Highest and lowest values of stock price of a day, volume of stock, simple moving average, and exponential moving average can be considered as the financial indicators in the technical analysis. These are more suitable as the input for machine learning methods than fundamental analysis [38, 39].

Due to the substantial increase of the computational power, machine learning has become popular stock market forecasting method [40]. Some of the prominent machine learning methods include support vector machines (SVM) and neural networks. Schumaker and Chen [41] used SVM to classify the direction (rise or drop) of future stock prices. Lin et al. [42] proposed a quasi-linear SVM method for stock market prediction, which selected the subset of financial indexes as the model's weighted inputs. Devi et al. [43] employed metaheuristic (cuckoo optimisation) method for training SVM parameters for stock market forecasting. Dase et al. [44] employed neural networks to improve stock market forecasting accuracy. Liao et al. [45] incorporated stochastic time effective function with neural networks and used different volatility parameters to assess the predictive performance of the model. Moghaddam et al. [46] employed neural networks for two type of input datasets in order to forecast the daily stocks of the NASDAQ stock exchange. Chopra et al. [47] subdivided nine stocks based on volatility and market capitalization and demonstrated that neural networks have good ability for stock price forecasting before and after demonetization in India.

Evolutionary optimisation methods such as genetic algorithms (GAs) have widely been been used to train neural networks and SVMs [48]. Khatibi et al. [49] presented combination of GAs with SVMs which used various financial indicators as input features that provided better performance when compared to neural networks alone. Qiu et al. [50] combined GAs with neural networks to enhance the accuracy of stock market forecasting index. Moreover, neural networks trained by hybrid of simulated annealing and GAs significantly enhanced the prediction accuracy over traditional backpropagation neural networks [51].

Furthermore, other hybrid methods that fall in the field of artificial intelligence and machine learning have also been used in stock market forecasting. Guresen et al. [52] presented a comparison amongst multilayer perceptron, dynamic neural networks, and hybrid neural networks that featured generalized autoregressive conditional heteroscedasticity (GARCH) model. The authors reported that multilayer perceptron provided best performance. Rathnayaka et al. [53] presented a hybrid model based on neural networks and autoregressive integrated moving average (ARIMA) for forecasting the Colombo stock exchange which provided better predictive ability under a high volatility than conventional time series forecasting methods. Zhong and Enke [54] applied three dimensionality reduction techniques which include principal component analysis (PCA), fuzzy robust principal component analysis

(FRPCA), and kernel-based principal component analysis (KPCA) with neural networks to estimate the daily direction of the future stock market returns. The authors reported that the combination of neural networks and PCA provides more accurate results when compared to the other combinations.

## 2.2 Neural networks for forecasting

Deep learning refers to a special type neural networks which consists of multiple processing layers and enables high-level abstraction to model data [55]. In the literature, deep learning commonly refers to the outstanding models such as recurrent networks (RNNs), convolutional neural networks (CNNs), deep belief networks, and long short-term memory networks (LSTMs) [56, 57]. In recent years, with more computing power and massive datasets, deep learning models have demonstrated excellent performance in different fields, such as sentiment analysis [58], image analysis [59] and natural language processing [60].

The main advantage of deep learning models is the ability to automatically extract the good features of input data through the general-purpose learning procedure [61]. Therefore, deep learning has also been widely used in various forecasting applications. Amarasinghe et al. [62] investigated the effectiveness of CNNs for individual building level energy load forecasting. Huang and Kuo [63] combined CNNs and LSTMs for air pollution (PM 2.5) forecasting. Sudriani et al. [64] utilized LSTMs for forecasting discharge level of Cimandiri River which was beneficial for managing water resources.

The financial community has received a boost in developing solutions with deep learning models for financial forecasting research. Ding et al. [65] utilized CNNs to evaluate the impact of different events on stock price behavior in the short, middle and long term. Nelson et al. [66] used LSTMs to forecast the future trends of stock market based on the price history and technical analysis indicators. Apart from these, innovative approaches for training conventional neural networks have been utilised. Chandra et al. [67] used co-evolutionary RNNs for stock market forecasting and proposed framework for mobile application.

Bayesian neural networks have strength in forecasting due to promising prediction accuracy with uncertainty quantification. Different Bayesian neural networks such as recursive Bayesian recurrent neural networks [68] and evolutionary MCMC Bayesian neural networks [69] have been used for time series forecasting. Liang et al. [70] proposed an MCMC algorithm for neural networks for selected time series problems. Chandra et al. [15] presented Langevin gradient Bayesian neural networks with parallel tempering MCMC, which used high-performance computing for time series prediction. Bayesian neural networks have been applied to various fields such as railway passenger flow [71], a certain index of the national economy [72], and short-term commodity prices [73]; these applications have reported promising forecasting performance.

## 2.3 COVID-19 impact on world economy

As mentioned earlier, the first phase of COVID-19 has a devastating impact on the world economy and the stock market. Ahmar et al. [74] presented a study for forecasting effect of COVID-19 on stock market in Spain using ARIMA model. The authors combined the ARIMA model with $\alpha$-Sutte indicator which uses 4 previous data points for forecasting and was more suitable when compared to ARIMA model alone. They also reported that the increase in the number of COVID-19 cases had direct effect on the stock market. Ali et al. [75] utilized the GARCH model to evaluate the volatility of the financial markets during the transfer of COVID-19 from China to Europe and then to the United States. The authors also

performed a bivariate regression between the returns and volatility of various financial securities during the first phase of COVID-19. The results show that China gradually stabilized, while the global market has experienced a sharp drop of financial security with the spread of the epidemic. In another study, the researchers tried to build a predictive model to assess the relationship between health-related news and stock returns in the worst-hit countries by COVID-19 [76].

Moreover, Maliszewska et al. [23] evaluated the effect of COVID-19 on gross domestic product (GDP) and trade in four major areas that included the reduction in employment rate and capital demand, higher costs in international trade, sharp decrease in international tourism, and declining demand for services that require close human interaction. The economic model which is conceptually similar to the approach of modelling severe acute respiratory syndrome (SARS) outbreak in 2002 was applied to approximate the potential impact of COVID-19 on the global economy [77]. Nicola et al. [78] analyzed in detail the socio-economic impact of the COVID-19 on various industries and focused on the primary industry related to raw material mining, secondary industry related to finished product production, and tertiary industry including service-providing industries. McKinbbin and Fernando [79] predicted seven different scenarios of how COVID-19 might evolve in the coming year, which indicated that the outbreak can have a significant impact on the global economy in the short term. Guan et al. [80] utilized the latest global trade modelling framework to analyze the supply-chain effects on a number of idealized lock-down scenarios.

## 3 Methodology

### 3.1 State-space reconstruction

State-space reconstruction refers to embedding a time series into a vector so that it can be trained by machine learning models [81]. According to the Taken's theorem, the embedding process must ensure that the original characteristics of the time series is retained [82]. Given a univariate time series, we can construct a multi-dimensional space vector by taking a point on the fixed delay of the original system. Using Taken's embedding theorem [82], the state-Space reconstruction is given as follows.

Suppose the actual series of closing stock price is $[x_1, x_2, \ldots, x_N]$, where $N$ is the length of the series. First, we choose the embedding dimension $m$ and a time lag $T$, and then capture windows of size $m$ denoted by vector $\bar{x}$ for every $T$ delay until $N$ is reached. Our problem is multi-step prediction where we have $n$ prediction horizons denoted by vector $\mathbf{y}$. The reconstructed vector by state-space embedding is denoted by $[\bar{x}, \mathbf{y}]$. Hence, for the first instance, we have

$$\bar{\mathbf{x}}_1 = [x_1, x_2, x_3, x_4, \ldots, x_m]$$

$$\mathbf{y}_1 = [x_{m+1}, x_{m+2}, x_{m+3}, \ldots, x_{m+n}]$$

In the same way, we can obtain the rest of the instances for the entire time series as given below.

$$\bar{\mathbf{x}}_t = [x_{1+(t-1)T}, x_{2+(t-1)T}, x_{3+(t-1)T}, x_{4+(t-1)T}, \ldots, x_{m+(t-1)T}]$$

$$\mathbf{y}_t = [x_{m+(t-1)T+1}, x_{m+(t-1)T+2}, x_{m+(t-1)T+3}, \ldots, x_{m+(t-1)T+n}]$$

## 3.2 Neural networks

Canonical neural networks which are also known as multi-layer perception employs multiple layers in the model, where each layer features neurons that propagate information for layers ahead a shown in Fig 1. Each neuron can receive the signal of the neuron in the previous layer and generate the output to the next layer. The first layer is called the input layer, the last layer is called the output layer, and the other intermediate layers are called hidden layers. The hidden layer could be one layer or feature multiple layers.

A neural network model $f(x)$ can be defined as a composition of other functions. Given a series of input-output pairs $\{\bar{\mathbf{x}}_t, \mathbf{y}_t\}$, the model is trained to approximate the function $f$ such that $f(\bar{\mathbf{x}}_t) = \mathbf{y}_t$ for all pairs. In our setting,

$$f(\bar{\mathbf{x}}_t) = g\left(\delta_o + \sum_{h=1}^{H} v_h \times g\left(\delta_h + \sum_{i=1}^{m} w_{ih}\bar{x}_{t,i}\right)\right), \tag{1}$$

where, $m$ is the input number and $H$ is the number of hidden layers. The function $g(.)$ is the sigmoid activation function which is used in the hidden and output layers. The setup for multi-step ahead time series prediction problem using neural networks with one hidden layer is shown in Fig 1. The complete set of parameters for the neural network model is shown in Fig 1 $\boldsymbol{\theta} = (\tilde{\mathbf{w}}, \tilde{\mathbf{v}}, \boldsymbol{\delta})$, where $\boldsymbol{\delta} = (\delta_o, \delta_h)$. $\tilde{\mathbf{w}}$ is the weight of the input to hidden layer. $\tilde{\mathbf{v}}$ is the weight of the hidden to output layer. $\boldsymbol{\delta}_h$ is the bias for the hidden layer, and $\boldsymbol{\delta}_o$ is the bias for the output layer.

Stochastic gradient descent (SGD) is one of the prominent methods of training neural networks. SGD is an iterative method to optimize a differentiable objective function with help of gradients [83]. In some high-dimensional optimization problems, SGD reduces the computational burden by achieving faster iterations with a lower convergence rate [83]. Training neural networks also can be considered as solving the non-convex optimization problem:

arg min $L(w)$, where $w \in R^n$ is the set of parameters and $L$ is the loss function. The iterations of SGD can be given as

$$w_k = w_{k-1} - a_{k-1}\nabla L(w_{k-1})$$

where, $w_k$ denotes the $k^{th}$ iteration, $a_k$ is the learning rate, and $L(w_k)$ denotes the gradient at $w_k$.

We note that the learning rate is user defined parameter which depends on the type of problem and typically it is determined in trial experiments. Hence, extension of the SGD consider adapting the learning rate automatically during the learning process. Adaptive moment estimation (Adam) is an effective stochastic optimization method that only requires first-order gradients with a small amount of memory requirement focused in adapting learning rate [84]. Adam calculates the individual adaptive learning rates of the parameters from the estimates of first and second moments of the gradients. The Adam-based weight update is expressed as follows

$$w_k = w_{k-1} - a_{k-1} \cdot \frac{\sqrt{1 - \beta_2^k}}{\sqrt{1 - \beta_1^k}} \cdot \frac{u_{k-1}}{\sqrt{n_{k-1}} + \epsilon}$$

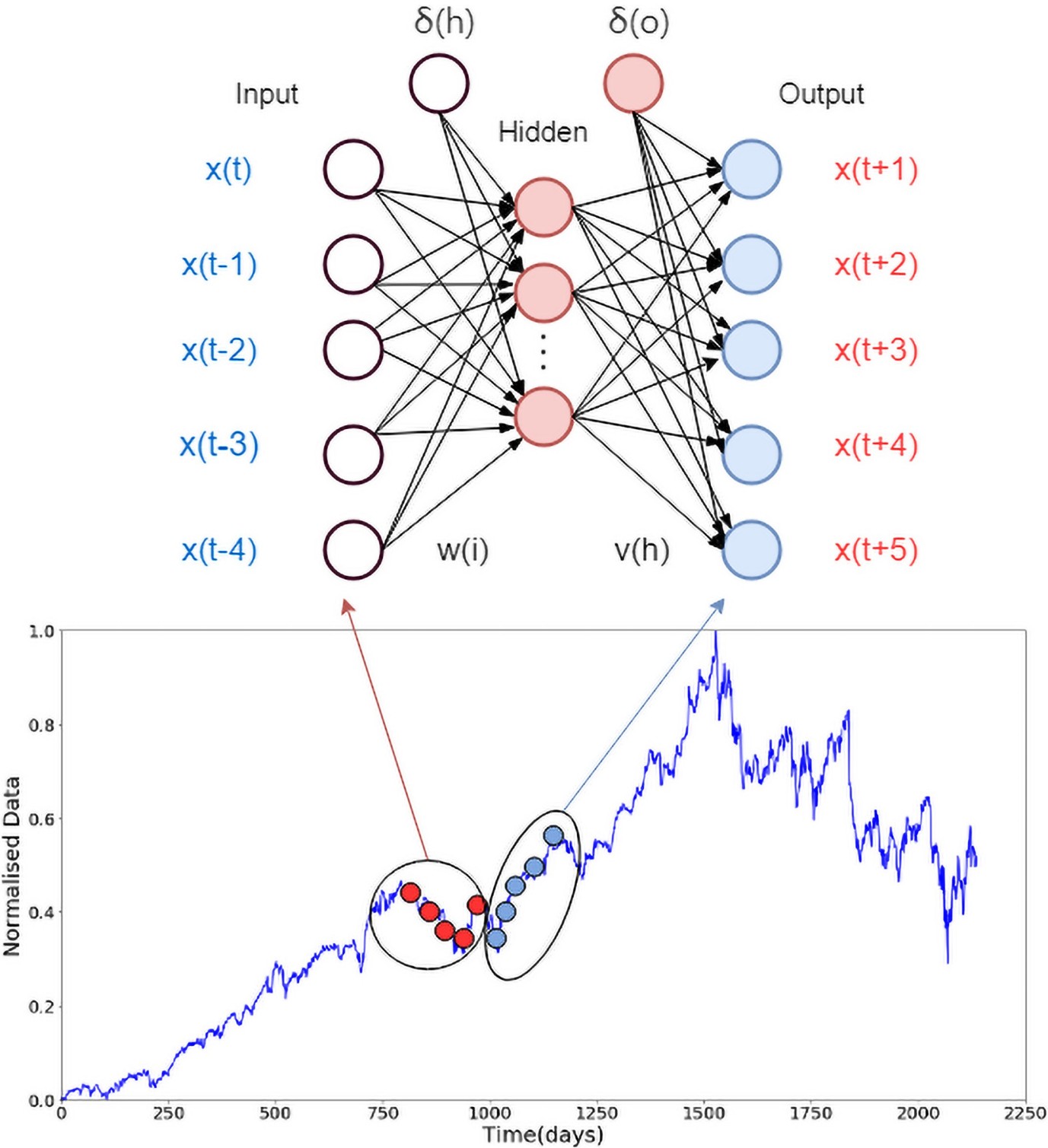

**Fig 1. The time series (shown in red circles) is used as input for the neural network which predicts 5 steps-ahead in time (shown by black circles).** A sliding window approach is used to reconstruct the dataset in this way using Taken's theorem.

where

$$u_{k-1} = \beta_1 u_{k-2} + (1 - \beta_1)\nabla f(w_{k-1}) \tag{8}$$

$$n_{k-1} = \beta_2 n_{k-2} + (1 - \beta_2)\nabla f(w_{k-1})^2$$

where, $\beta_1$ and $\beta_2$ are first and second moment estimates, respectively. $\epsilon$ is a small scalar used to prevent division by 0.

### 3.3 Bayesian neural networks

Bayesian neural networks provide a probabilistic implementation of a standard neural network with the key difference where the weights and biases are represented via posterior probability distributions rather than single point estimates [85, 86] as shown in Fig 2. Similar to standard neural networks, Bayesian neural networks also have universal continuous function approximation capabilities.

The challenge of Bayesian inference is in sampling to approximate (learn) the posterior distribution of neural network weights and biases. The inference procedure begins by setting prior distributions over the weights and biases. Then the sampling scheme (such as MCMC) employs a likelihood function that takes into account the training data accepting or rejecting a proposed sample. The implementation of Bayesian neural network using MCMC sampling is shown in Fig 2. Due to non-linear activation functions in the Bayesian neural network, the conjugacy of prior and posterior is lost. Moreover, due to large number of parameters given different applications, it is difficult to get informative priors and hence, it is challenging is to sample the posterior distribution.

We can construct the likelihood function using the set of weights and biases (Eq (1))) $\boldsymbol{\theta}$, for $M$ network parameters and $S$ training instances. We note that we use a signal plus noise model where an additional parameter ($\tau^2$) is used to cater for the noise. Hence, for model output

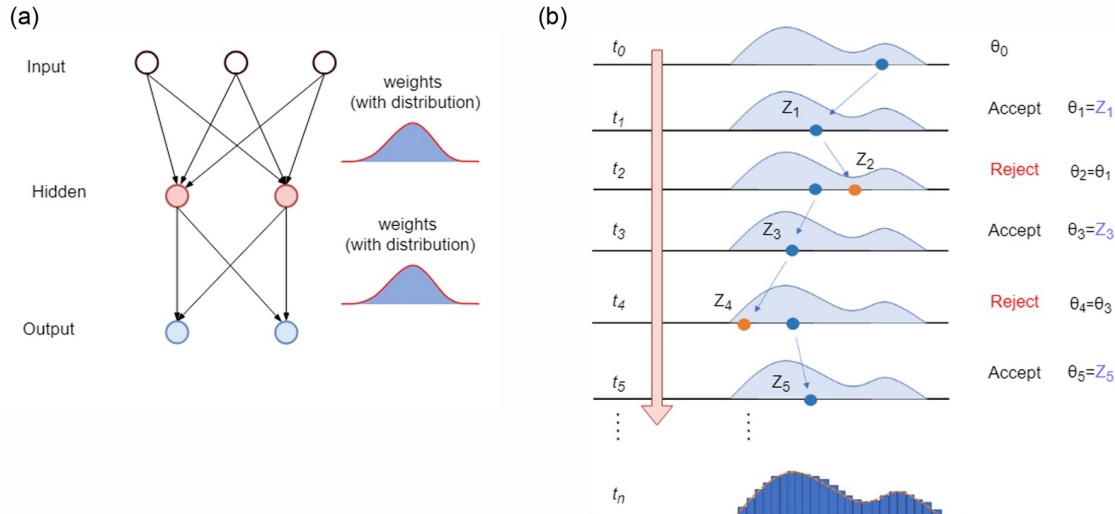

**Fig 2. Bayesian neural network and MCMC sampling.** Note that the posterior distribution is shown that represents weights in Panel (a).

$f(\bar{\mathbf{x}}_\mathbf{t})$ with given input features $\bar{\mathbf{x}}_\mathbf{t}$, we have

$$
\begin{aligned}
p(\mathbf{y_s}|\boldsymbol{\theta}) &= -\frac{1}{(2\pi\tau^2)^{S/2}} \times \\
&\quad \exp\left(-\frac{1}{2\tau^2}\sum_{t\in S}(\mathbf{y_t}-f(\bar{\mathbf{x}}_\mathbf{t}))^2\right)
\end{aligned}
\tag{2}
$$

which satisfies the multivariate probability density function.

The prior is based on Gaussian distribution in the case of $\boldsymbol{\theta}$ and Gamma distribution in the case of $\tau^2$ as shown below.

$$
p(\boldsymbol{\theta}) \quad \propto \quad \frac{1}{(2\pi\sigma^2)^{L/2}} \times \exp\left\{-\frac{1}{2\sigma^2}\left(\sum_{i=1}^{M}\theta\right)\right\} \times \tau^{2(1+v_1)}\exp\left(\frac{-v_2}{\tau^2}\right)
\tag{3}
$$

where, $\sigma^2$ is determined by exploring variance in weights and biases of trained neural networks for similar applications. Moreover, $v_1$ and $v_2$ are user defined constants.

## 3.4 Sampling using parallel tempering MCMC

Parallel tempering MCMC features an ensemble of replica samplers that can run in parallel and has the ability to sample multi-modal posterior distributions [87, 88]. A user defined temperature ladder corresponds to every replica in the ensemble, where the higher temperature values have higher probability to accept weaker proposals. The ensemble is defined by a total of $R$ replicas at temperature level $T_m$ specified by

$$
\Theta = (\theta^{[1]}, \ldots, \theta^{[R]})
$$

where $m$ denotes the replica. We sample $\theta$ from the posterior distribution by proposing $\theta^p$ from a known distribution $q(\theta)$. Given the proposed value $\theta^p$, the chain moves with a probability $\alpha$ or remains at its current position $\theta^k$. We note that $\alpha$ is chosen to ensure that the chain is reversible and has stationary distribution $p(\theta|\mathbf{D})$ given data $\mathbf{D}$.

Algorithm 1 provides further details where parallel tempering MCMC is used for global exploration which then transforms to canonical MCMC via parallel computing for local exploration. The transformation is done by changing the temperature ladder to series of 1's. The local exploration also features exchange of neighboring replicas. The user needs to set the percentage of samples for global exploration phase in advance along with hyper-parameters like the maximum number of samples ($Max_{samples}$), and the swap interval ($Swap_{int}$) which is typically set to a few iterations to support efficient inter-process communications in parallel computing environment as shown in Fig 3. After every few iterations (ad defined by $Swap_{int}$), the algorithm determines if the neighboring replicas require to be swapped which is necessary to improve the efficiency of exploring the posterior distribution. The replica swap is determined by the Metropolis-Hasting acceptance criterion which is similar to within replica transition. After the termination condition is met, the respective replica posterior distributions are combined after discarding the burn-in period which is marked by the parallel tempering MCMC global exploration phase. In this way, we ensure that only the true posterior (replica's with temperature of 1's) are part of the posterior distribution.

We use Langevin-gradient proposal distribution [89] that essentially features a one-step gradient over Gaussian noise. At a given chain position $k$, our new proposal $\theta^p$ is given as

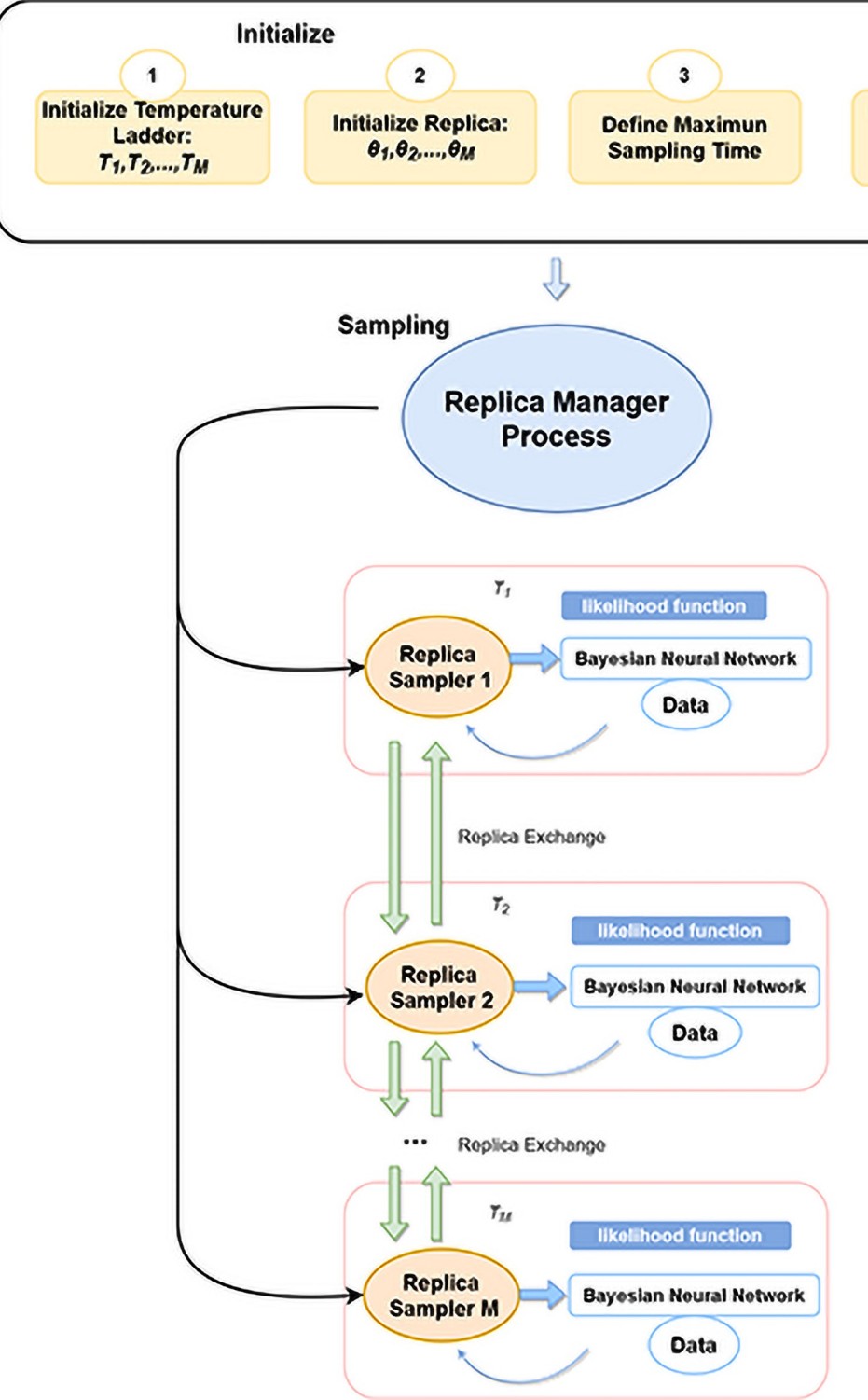

**Fig 3. An overview of the different replicas that are executed on a parallel computing architecture.**

follows

$$\boldsymbol{\theta}^p \quad \sim \quad \mathcal{N}(\bar{\boldsymbol{\theta}}^k, \Sigma_{\boldsymbol{\theta}}), \text{ where} \tag{4}$$

$$\bar{\boldsymbol{\theta}}^k \quad = \quad \boldsymbol{\theta}^k + r \times \nabla E(\boldsymbol{\theta}^k), \tag{5}$$

$$E \quad = \quad \sum_{i \in S}(\boldsymbol{y}_i - f(\bar{\boldsymbol{x}}_i))^2 \tag{6}$$

$$\nabla E(\boldsymbol{\theta}^{[k]}) \quad = \quad \left(\frac{\partial E}{\partial \theta_1}, \cdots \frac{\partial E}{\partial \theta_L}\right) \tag{7}$$

where $r$ is the learning rate, $\Sigma_{\theta} = \sigma_{\theta}^2 I$, $I$ is the $L \times L$ identity matrix, $\bar{x}$ is the univariate time series input data vector (window) denoted by $i$ for $S$ data instances, $\boldsymbol{y}_i$ is the time series data vector for $h$ prediction horizons, and $L$ refers to the total number of model parameters (weights and biases). Based on a user defined probability $\phi$, the proposal $\boldsymbol{\theta}^p$ can be either

- A one-step gradient descent based weight update known as Langevin-gradient (LG) proposal distribution (Eq 4),

- A random-walk (RW) proposal distribution where Gaussian noise from distribution centered at mean of 0 and variance, $\mathcal{N}(0, \Sigma_{\theta})$.

We note that the ensemble of replicas execute in parallel that have stationary distributions which are equal to up to a proportionality constant defined by the temperature ladder, $p(\theta|\mathbf{D})^{\beta}$; where $\beta \in [0, 1]$ corresponds to the temperature ladder that features geometric spacing. $\beta = 0$ corresponds to a uniform stationary distribution and $\beta = 1$ refers to the posterior. Hence, the replicas that feature smaller temperature levels from $\beta$ can provide global exploration, while those with higher values provide local search or exploitation. For each replica $m$ in the ensemble, we propose $\theta_m^p$ using Langevin-gradient conditional on current value $\theta_m^k$, that is $\theta_m^k \sim q(\theta|\theta_m^k)$. The within replica transition determines if the proposed value of $\theta_m^p$ remains at its original location $\theta_m^k$ or gets updated by a probability as given

$$\alpha = min\left(1, \frac{p(\theta_m^p|D)^{\beta_m} q(\theta_m^k|\theta_m^p)}{p(\theta_m^k|D)^{\beta_m} q(\theta_m^p|\theta_m^k)}\right). \tag{8}$$

The proposal becomes part of the posterior distribution once it is accepted.

**Algorithm 1**: Langevin-gradient Parallel Tempering MCMC

**Result**: Draw samples from distribution $p(\theta|\mathbf{D})$
i. Define maximum number of samples: $Max_{samples}$, swap interval: $Swap_{int}$ and number of replicas: $R$
ii. Initialize $\theta_m = \theta_m^{[0]}$ for each replica $m$
iii. Define the Langevin-gradient probability $\phi$
iv. Define percentage of samples for global exploration
v. Define the temperature ladder that uses geometric spacing, $\beta$.
**while** $Max_{samples}$ **do**
 **for** $m = 1, ..., R$ **do**
 **for** $k = 1, ..., Swap_{int}$ **do**
 **Step 1: Within Replica sampling**
 1.1 Propose new sample (solution)
 Draw $l \sim U[0, 1]$
 **if** $((l < \phi)$ **then**
 Get $\theta_m^p$ using Langevin-gradient proposal distribution (Eq 4)

```
   else
      Get θᵖₘ using Random-Walk proposal distribution (𝒩(0, Σ_θ))
   end
   1.2 Compute acceptance probability α (Eq 8)
   1.3 Acceptance criterion Draw u ~ U[0, 1]
   if u < α then
      θᵏₘ = θᵖₘ
   else
      θᵏₘ = θᵏ⁻¹ₘ
   end
   end
  end
end
Step 2: Replica exchange
2.1.Compute acceptance probability for neighboring replicas
2.2. Exchange neighboring replica if accepted.
*Local/Global exploration phase
if global is true then
   βₘ = βₘ
else
   βₘ = 1
end
end
```

## 4 Experiments and results

In this section, we provide details about the datasets and present research design with computational results.

### 4.1 Data

We choose 4 stocks from 4 different countries to evaluate the performance of the respective methods. These include 3M Company(MMM) from United States, China Spacesat Company Limited (600118.SS), Commonwealth Bank of Australia (CBA.AX), and Daimler AG (DAI. DE) from Germany. The respective datasets feature the closing price with the time period from 01/01/2012 to 01/07/2020 from Yahoo Finance. MMM is an American world-renowned multinational company with diversified products, which cover various fields such as household goods and medical supplies. China Spacesat Company Limited is an aerospace high-tech enterprise specializing in the development and application of small satellites in China. Commonwealth Bank is the largest commercial bank in Australia. Daimler AG is a German company which is the largest commercial vehicle manufacturer and the largest luxury car manufacturer in the world. In the case of China Spacesat Company Limited, the data after January 2020 is affected by COVID-19. In the case of the remaining companies, the data after March 2020 is affected by COVID-19.

We apply data normalization to the original data time series data to a boundary of [0, 1] using min-max normalization given as

$$x'_i = \frac{x_i - x_{min}}{x_{max} - x_{min}},$$ (9)

where, $x$ is the adjusted closing price time series.

### 4.2 Experiment setup

In parallel tempering MCMC, we use burn-in rate of 0.5 with maximum of 100,000 samples with maximum temperature value of 2. Note that the burn-in rate also defines the first-phase

where global search is enforced by parallel tempering in of Algorithm 1. We use 10 replicas with swap interval is 5. The Langevin-gradient proposals use learning rate of 0.1 and applied with a probability of 0.5. We use one hidden layer feedforward neural network with 5 output units where each output unit denotes a prediction horizon for the 5-step-ahead stock black-price prediction problems. We apply Taken's embedding theorem to reconstruct data with dimension D = 5 and time-lag T = 2. Hence, 5 input neurons with 5 hidden neurons are used in the respective neural network models.

The experiments are executed as follows.

- Evaluate multi-step-ahead stock price prediction using novel Bayesian neural networks before COVID-19.

- Compare the results with feedforward neural network using Adam optimiser (FNN-Adam) and feedforward neural network using stochastic gradient descent (FNN-SGD) training algorithm.

- Evaluate multi-step-ahead stock price prediction using novel Bayesian neural networks during COVID-19.

The results report the mean and 95% confidence interval from 30 experimental runs with different weight and bias initialisation in the respective models for 5-step ahead prediction. The prediction performance is measured by root mean squared error(RMSE) given as follows.

$$RMSE = \sqrt{\frac{1}{N}\sum_{i=1}^{N}(y_i - \hat{y}_i)^2} \tag{10}$$

where, $y_i$ is the actual value and $\hat{y}_i$ is the predicted value, and N is the number of data-points for a single prediction horizon (step).} We use RMSE since it is one of the key performance measures for time series forecasting in the literature. Other measures such as Mean Absolute Error (MAE), and Normalised Mean Squared Error (NMSE) can also be used. Note that in our past work [90], we found that NMSE for example, does not change the key conclusions for time series problems, hence we used RMSE only.

## 4.3 Prediction results pre-COVID-19

The first 80% of the data is used for training and remaining is used for testing. Table 1 gives further details of the time frame considered and indicates the exact dates for the respective stocks.

Fig 4 reports the performance of the respective method in prediction of future trends of given stock prices that include MMM, 600118.SS, CBA.AX and DAI.DE, respectively. The results show the prediction horizon (step), the mean RMSE with a 95% confidence interval as error-bars. Table 2 further presents the results numerically.

In the case of MMM (Fig 4(a)), Bayesian neural network (Bayes-FNN) performs the best with the lowest RMSE compared to FNN-Adam and FNN-SGD where the error increases with the prediction horizon. Fig 4(b) presents results for stock 600118.SS that show that Bayes-FNN

**Table 1. Time span of data considered for each stock-price pre-COVID-19.**

|  | MMM | 600118.SS | CBA.AX | DAI.DE |
|---|---|---|---|---|
| Train (80%) | 3.1.2012-24.5.2018 | 4.1.2012-28.5.2018 | 3.1.2012-24.7.2018 | 2.1.2012-22.5.2018 |
| Test (20%) | 25.5.2018-31.12.2019 | 29.5.2018-31.12.2019 | 25.7.2018-31.12.2019 | 23.5.2018-30.12.2019 |

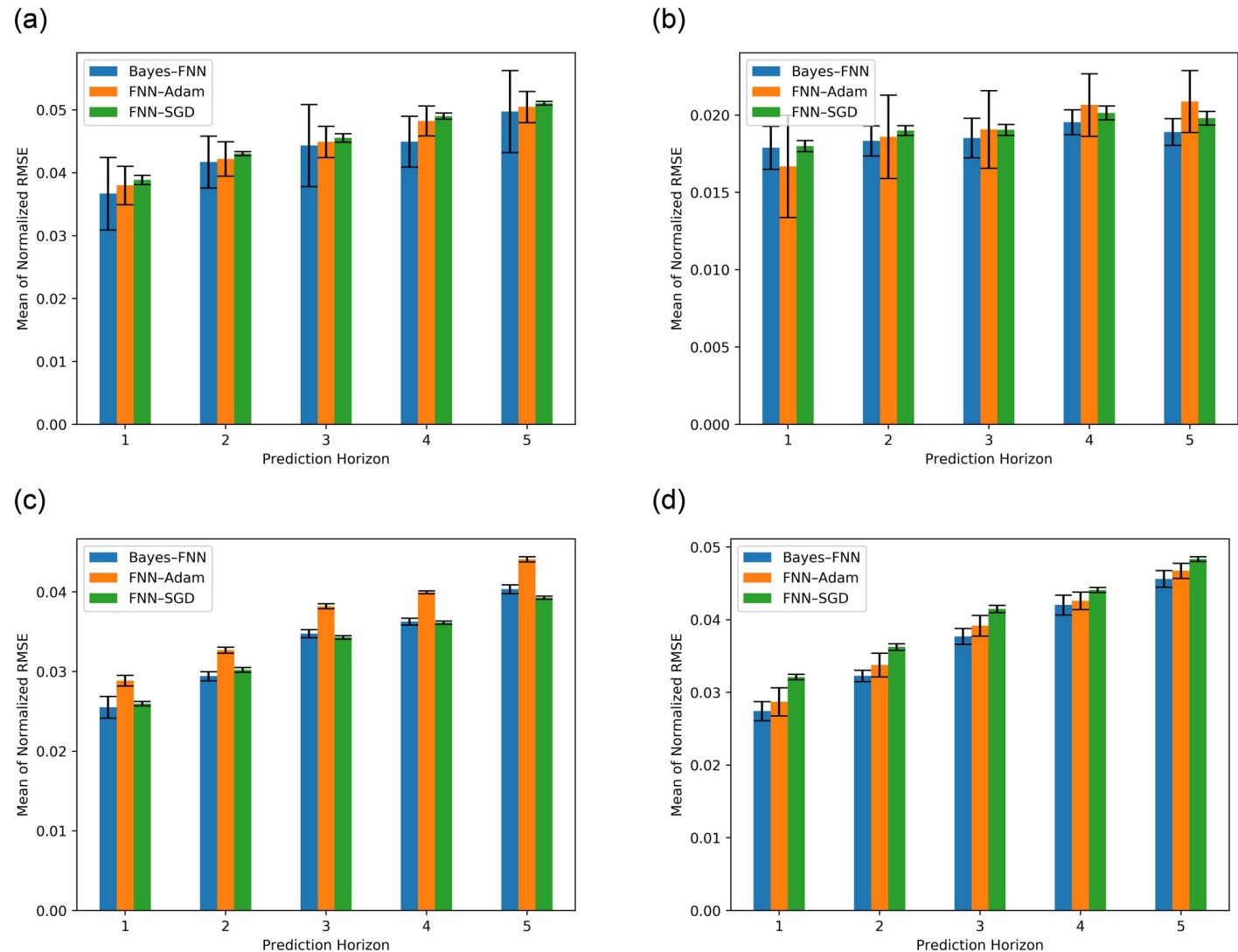

**Fig 4. Prediction performance (mean of normalized RMSE) for the three models (Bayes-FNN, FNN-Adam, and FNN-SGD).** Note that the predictions were normalized in range of [0, 1].

gives best performance in all the prediction horizons, expect for 1. The error increases with the prediction horizon. In Fig 4(c), Bayes-FNN shows the best performance in the prediction horizon 1 and 2, which is overtaken by FNN-Adam in the rest, while the RMSE increases with the prediction horizon. In the case for stock DAI.DE, Fig 4(d) shows that Bayes-FNN gives best performance and the RMSE increases as the prediction horizon increases.

Fig 5 (stock MMM), Fig 6 (stock 600118.SS), Fig 7 (stock CBA.AX), and Fig 8 (stock DAI. DE) show the prediction on the test dataset for the prediction horizons 1, 2 and 5 using Bayes-FNN. We notice that the uncertainty (shaded) is relatively small for stock DAI.DE and CBA. AX when compared to stock MMM and 600118.SS. We notice that Bayes-FNN gives very accurate predictions for horizon 1 and 2 in case of stock DAI.DE and CBA.AX and hence, there is lower uncertainty. In case of stock 600118.SS, there is similar level of accuracy in prediction horizons 1 and 2 but higher uncertainty while the stock MMM shows poor prediction and uncertainty quantification in the first half (less than 100 days).

**Table 2. Multi-step-ahead prediction (RMSE).**

| Problem | Step | Bayes-FNN | FNN-Adam | FNN-SGD |
|---|---|---|---|---|
| MMM | 1 | 0.03669±0.00577 | 0.03800±0.00305 | 0.03889±0.00073 |
| | 2 | 0.04173±0.00413 | 0.04221±0.00274 | 0.04305±0.00032 |
| | 3 | 0.04434±0.00654 | 0.04491±0.00249 | 0.04555±0.00067 |
| | 4 | 0.04495±0.00404 | 0.04826±0.00237 | 0.04902±0.00050 |
| | 5 | 0.04975±0.00652 | 0.05047±0.00249 | 0.05107±0.00031 |
| 600118.SS | 1 | 0.01789±0.00139 | 0.01669±0.00331 | 0.01800±0.00036 |
| | 2 | 0.01833±0.00098 | 0.01860±0.00269 | 0.01900±0.00032 |
| | 3 | 0.01852±0.00128 | 0.01908±0.00251 | 0.01904±0.00035 |
| | 4 | 0.01954±0.00081 | 0.02066±0.00202 | 0.02015±0.00045 |
| | 5 | 0.01891±0.00087 | 0.02088±0.00200 | 0.01980±0.00044 |
| CBA.AX | 1 | 0.02551±0.00136 | 0.02886±0.00067 | 0.02597±0.00028 |
| | 2 | 0.02942±0.00058 | 0.03270±0.00037 | 0.03021±0.00031 |
| | 3 | 0.03475±0.00050 | 0.03820±0.00031 | 0.03429±0.00021 |
| | 4 | 0.03628±0.00043 | 0.03996±0.00017 | 0.03616±0.00020 |
| | 5 | 0.04034±0.00055 | 0.04409±0.00033 | 0.03929±0.00020 |
| DAI.DE | 1 | 0.02743±0.00131 | 0.02871±0.00194 | 0.03212±0.00037 |
| | 2 | 0.03228±0.00078 | 0.03378±0.00163 | 0.03626±0.00046 |
| | 3 | 0.03771±0.00109 | 0.03918±0.00142 | 0.04150±0.00050 |
| | 4 | 0.04204±0.00136 | 0.04261±0.00121 | 0.04410±0.00035 |
| | 5 | 0.04562±0.00115 | 0.04674±0.00104 | 0.04836±0.00033 |

## 4.4 Results during COVID-19

Next, we apply Taken's embedding theorem to reconstruct data with dimension D = 5 and time-lag T = 1. The previous section featured data (training and test set) before the COVID-19 pandemic. In certain stocks, we included January and February 2020 data for training since COVID-19 was not widespread in countries such as USA, Germany and Australia during that period. We provide an investigation to check how the stock price changes during COVID-19 and effect of the stock price trend before COVID-19 on the stock price during COVID-19. Hence, we use the data previous of COVID-19 pandemic for different stocks as training data to set the model performance during COVID-19 and refer to it as Setup-1. Table 3 gives details of the dates considered for the training and test dataset for the respective stocks. In Setup-2, we include parts of the data during COVID-19 in the training set with all the training data from Setup-1 where the exact dates are given in Table 3. In general, we appended Setup-1 with data from March and April, 2020 that covers the first phase of the pandemic in the respective countries that affected the stocks. The major reason for doing this is to ensure that the training dataset covers the stock trend during the pandemic. Hence, our test datasets of both setups are different. We have entire COVID-19 dataset used as test dataset for Setup-1, while second half of COVID-19 data is used as test dataset in Setup-2. In this case, we only provide results using the Bayes-FNN method.

Fig 9 illustrates the performance of the Bayes-FNN method in forecasting the four stocks mentioned before respectively. Setup-1 (left) and Setup-2 (right) bar charts show the prediction horizon and the mean RMSE with a 95% confidence interval. Setup-1 features prediction during the whole COVID-19 period, and Setup-2 features predictions for the second half of the COVID-19. In general, RMSE for all the given stocks improved by reducing for Setup-2. Fig 9(b)) shows that Setup-2 gives a better performance with much lower error (RMSE) when

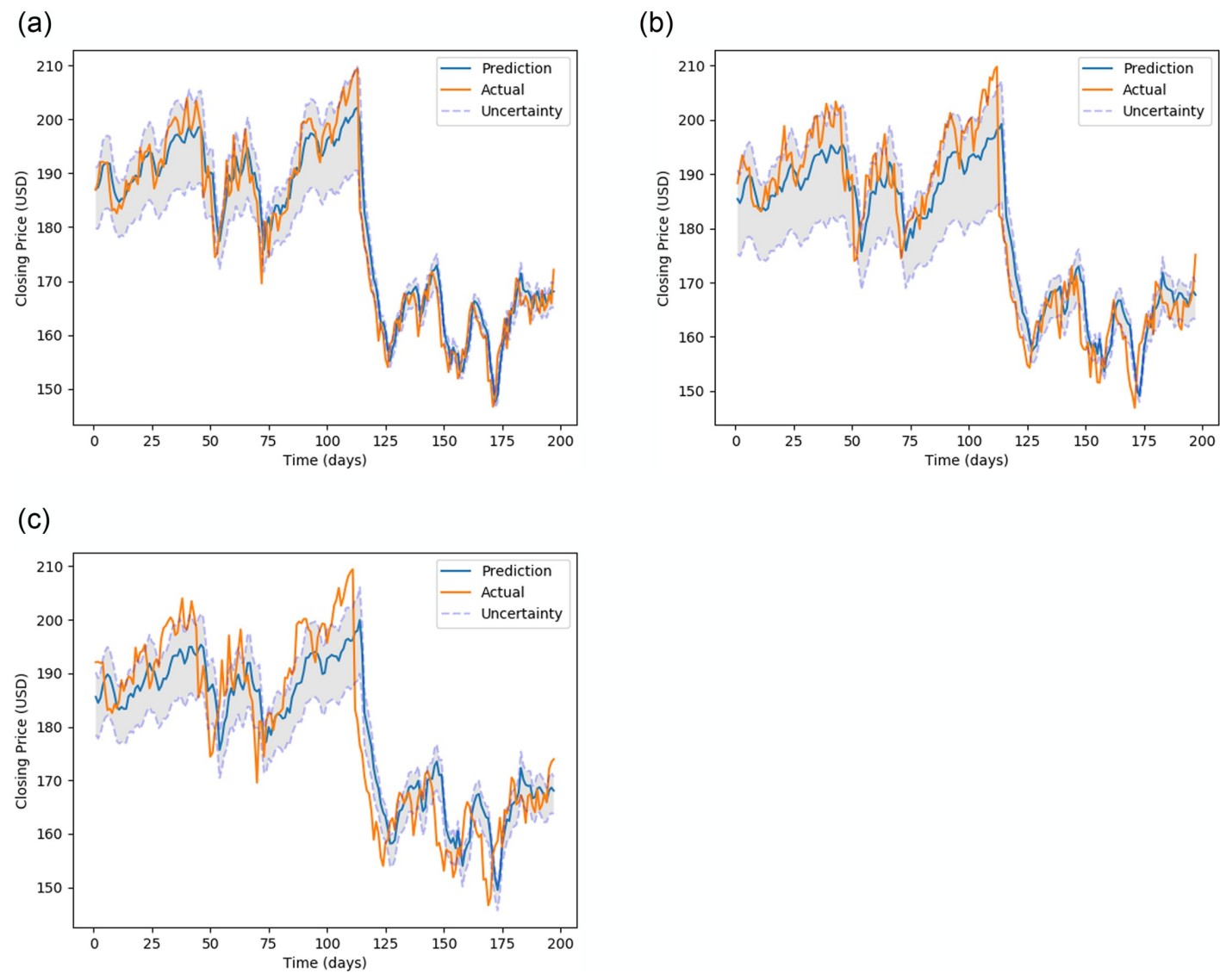

**Fig 5. Prediction and uncertainty (shaded region) for test data of stock MMM.**

compared to Setup-1 in Fig 9(a). Generally the error get larger with the prediction horizon during Setup 1, but there is not a clear trend when compared to Setup-2. In the case of stock MMM for Setup-1, prediction horizon 3 has the lowest error, while stock CBA.AX and DAI. DE show the best performance in the prediction horizon 1. In the case of stock 600118.SS, the performance in the prediction horizon 2 is the best. Given that we add the first half of time period affected by COVID-19 into the training set, the error during Setup-2 significantly decreases, but become larger with the prediction horizon which is natural for multi-step ahead prediction. Among the respective stocks, the prediction of stock 600118.SS performs the best for Setup-2. Table 4 further reports the results numerically.

Fig 10 (stock MMM), Fig 11 (stock 600118.SS), Fig 12 (stock CBA.AX) and Fig 13 (stock DAI.DE) show the prediction on test dataset for the prediction horizons 1, 2 and 5 using Bayes-FNN with Setup-1 and Setup-2. We notice that the uncertainty (shaded) is lower for

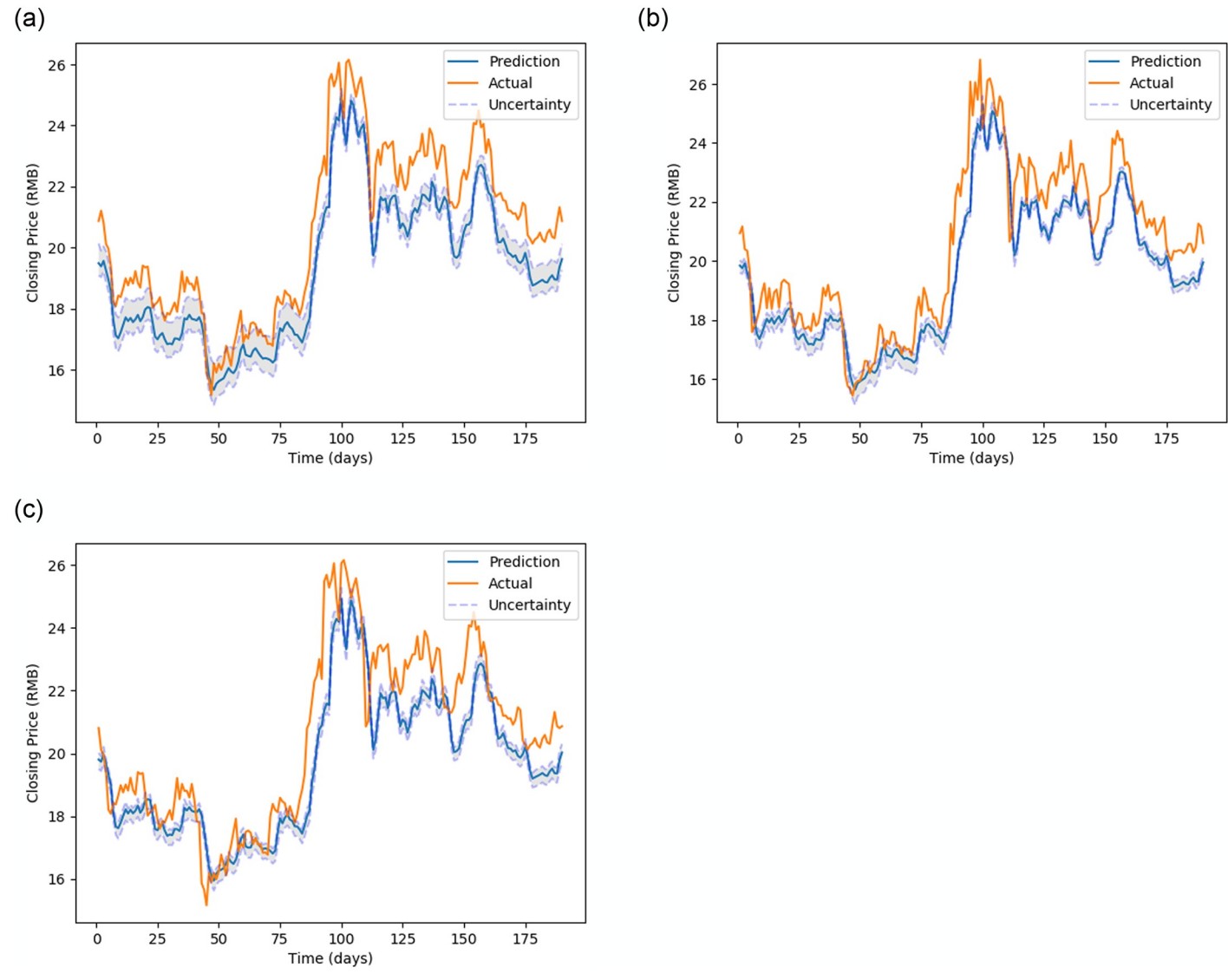

**Fig 6. Prediction and uncertainty (shaded region) for test data of stock 600118.SS.**

stocks DAI.DE and MMM for Setup 2 when compared to Setup 1. It is visually clear that the prediction in Setup-2 is generally better when compared to Setup-1.

The monthly volatility for the respective stock is presented with the red shaded area featuring the period during outbreak of COVID-19 (Figs 14 and 15). We measure volatility by the variance between returns from the particular stock index [91] and assume that there is 21 trading days per month. The respective stocks show high volatility in general during COVID-19. In Fig 14 after certain weeks (Fig 14(a) and 14(c)). In the other two stocks (Fig 15(a) and 15(c)), we notice a sharper drop sharply which are bank and luxury goods companies.

## 5 Discussion

The results show that prior to outbreak of COVID-19, Bayes-FNN provides one of the best performance which would be due to global-local exploration features of parallel tempering

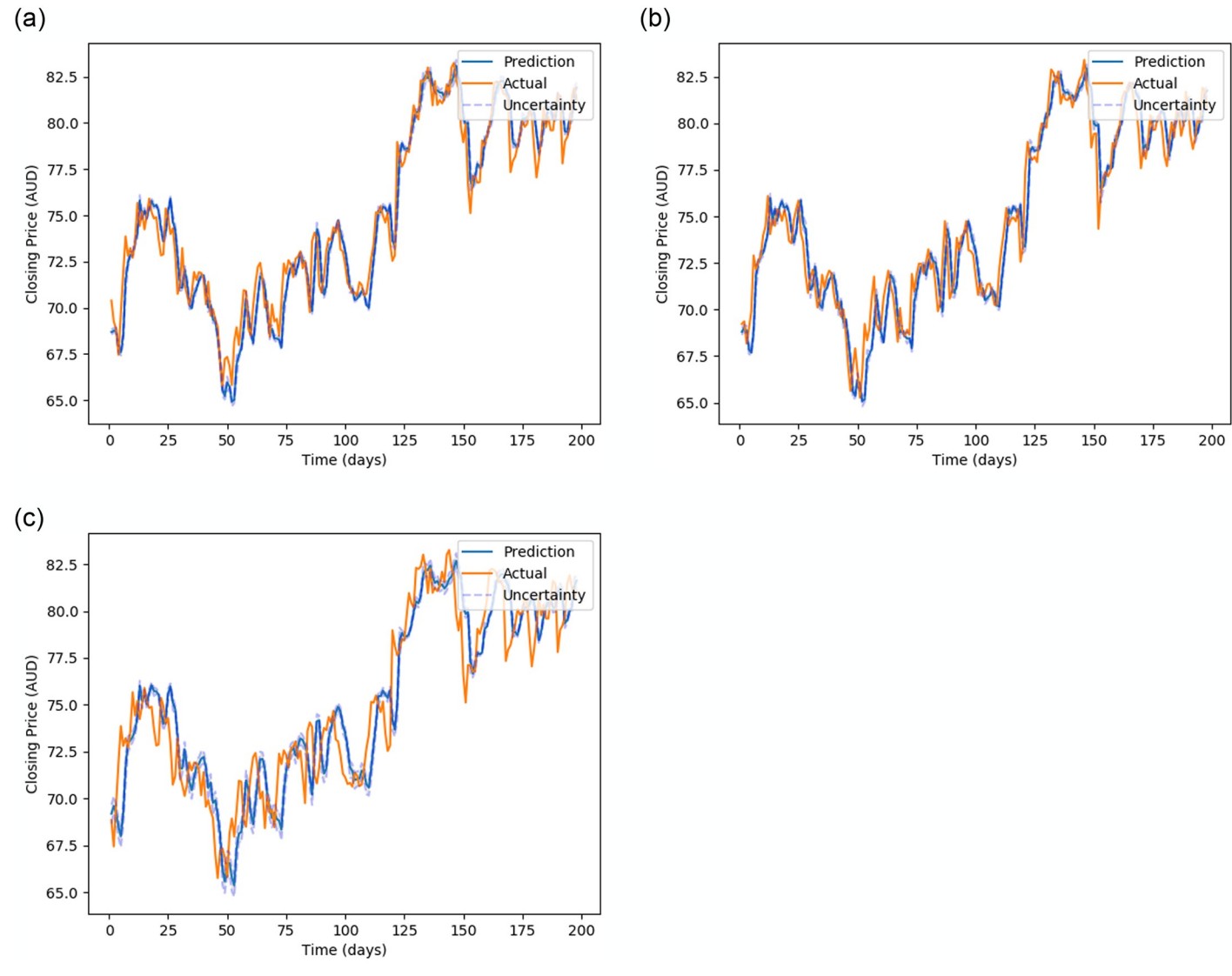

(a)

(b)

(c)

**Fig 7. Prediction and uncertainty (shaded region) for test data of stock CBA.AX.**

MCMC taking into account Langevin-gradient proposals. The accuracy decreased significantly as the prediction horizon increased. This is not surprising when taking into account other time series problems from in the literature by Chandra et al. [92] for one-step ahead prediction. Our work extended Bayes-FNN using parallel tempering MCMC used for one-step ahead prediction into multi-step ahead prediction for a challenging problem of stock price forecasting before and during COVID-19. We learned that Bayes-FNN scales well for multi-ahead prediction and provides better or competing performance when compared to state-of-art method (FNN-Adam). We note that typically stock price prediction does not have such a high level of uncertainty as weather predictions [93], however the situation during COVID-19 has significantly changed as shown by the volatility in Fig 14. The investors need to have confidence in the particular stock and rigorous uncertainty quantification in prediction. We know that good prediction accuracy is needed not just for the day ahead, but the behaviour of the stock days ahead is also of interest. A number of automated forecasting models are part of stock markets

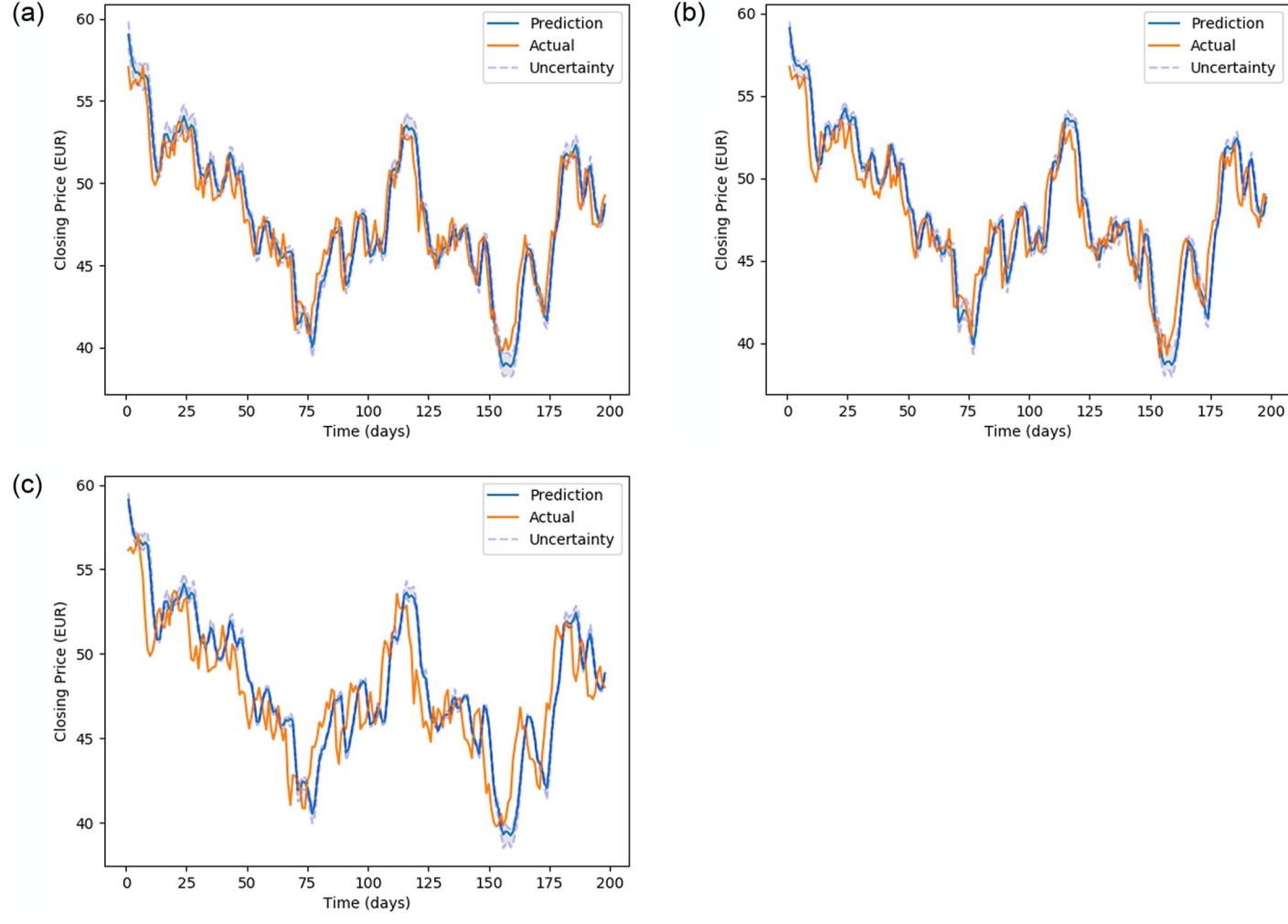

**Fig 8. Prediction and uncertainty (shaded region) for test data of stock DAI.DE.**

[94] and areas of machine learning such as reinforcement learning has made such automation possible [95]. Providing uncertainty quantification in automated stock forecasting models would give information to investors for better decision-making.

At the beginning of the COVID-19 pandemic, the stocks of various countries were greatly affected [96] which is quite clear from monthly volatility visualisation in Fig 14. We note that volatility statistical measure of the dispersion of returns for a stock where the higher the volatility implies riskier security [97]. The volatility also refers to uncertainty or risk related to the

**Table 3. Timespan considered for respective stocks during COVID-19.**

| Data | | MMM | 600118.SS | CBA.AX | DAI.DE |
|---|---|---|---|---|---|
| Setup 1 | Train (80%) | 26.10.2018-28.2.2020 | 31.1.2018-31.12.2019 | 31.10.2018-28.2.2020 | 7.11.2018-28.2.2020 |
| | Test (20%) | 2.3.2020-30.6.2020 | 2.1.2020-29.6.2020 | 2.3.2020-30.6.2020 | 2.3.2020-29.6.2020 |
| Setup 2 | Train (80%) | 6.9.2019-30.4.2020 | 16.4.2019-31.3.2020 | 2.9.2019-30.4.2020 | 10.9.2019-30.4.2020 |
| | Test (20%) | 1.5.2020-30.6.2020 | 1.4.2020-29.6.2020 | 1.5.2020-30.6.2020 | 4.5.2020-29.6.2020 |

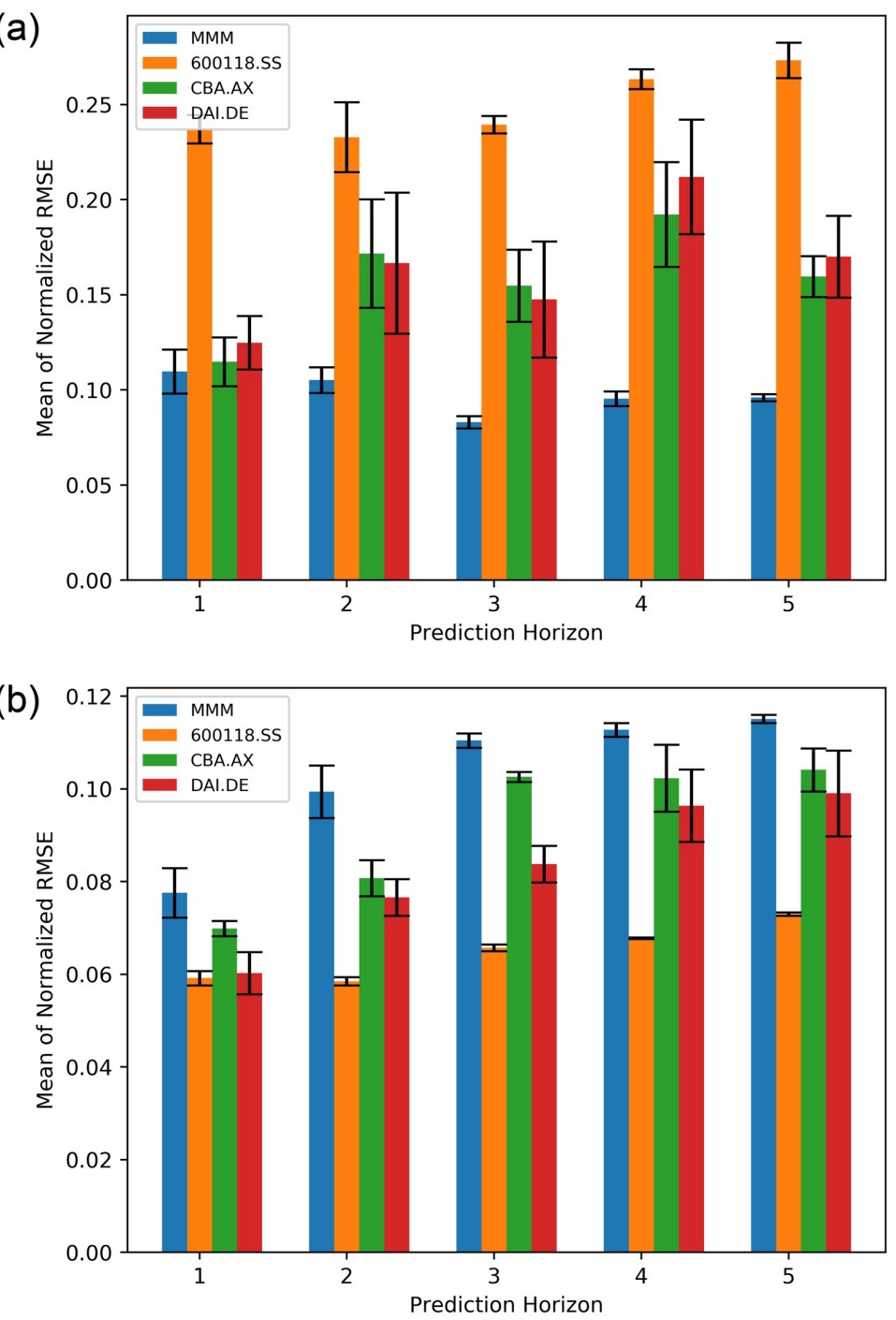

**Fig 9. Prediction performance (mean of normalized posterior RMSE) of respective stocks during COVID-19 using Bayes-FNN (Setup 1 vs Setup 2).** Note that the predictions were normalized in range of [0, 1].

changes in the stock [97]. The results during COVID-19 show that the uncertainty in the stock price is much higher due to the high volatility in the stock price. After training the model by adding part of data featuring high volatility (during COVID-19), the prediction accuracy of the model is greatly improved.

**Table 4. Multi-step-ahead prediction (RMSE) during COVID-19.**

| Data | Step | MMM | 600118.SS | CBA.AX | DAI.DE |
|---|---|---|---|---|---|
| Setup 1 | 1 | 0.10962±0.01153 | 0.23707±0.00748 | 0.11478±0.01281 | 0.12481±0.01401 |
| | 2 | 0.10516±0.00673 | 0.23278±0.01837 | 0.17165±0.02852 | 0.16664±0.03704 |
| | 3 | 0.08305±0.00319 | 0.23936±0.00457 | 0.15478±0.01897 | 0.14751±0.03054 |
| | 4 | 0.09534±0.00385 | 0.26333±0.00517 | 0.19222±0.02748 | 0.21195±0.03008 |
| | 5 | 0.09592±0.00186 | 0.27321±0.00933 | 0.15961±0.01077 | 0.16994±0.02151 |
| Setup 2 | 1 | 0.07758±0.00533 | 0.05916±0.00153 | 0.06984±0.00165 | 0.06024±0.00458 |
| | 2 | 0.09938±0.00567 | 0.05846±0.00089 | 0.08072±0.00391 | 0.07658±0.00394 |
| | 3 | 0.11044±0.00154 | 0.06571±0.00075 | 0.10260±0.00108 | 0.08375±0.00398 |
| | 4 | 0.11275±0.00149 | 0.06779±0.00018 | 0.10231±0.00723 | 0.09637±0.00783 |
| | 5 | 0.11509±0.00090 | 0.07300±0.00035 | 0.10410±0.00465 | 0.09903±0.00927 |

We approached the problem purely as a univariate time series forecasting; however, there is scope for taking a multi-variate approach in future study. Future work could consider empirical study of factors that affect the stock price for building model with multivariate forecasting approach during COVID-19. Some of the factors that greatly affect the stock market are the level of infections a country or region [98] has and the level of lock downs [99], which need to be incorporated in a comprehensive forecasting model. Furthermore, other models such as deep learning with LSTM network models [100] and convolutional neural networks [101] could further improve the prediction performance. Furthermore, there is also scope for Bayesian graph convolutional neural networks to capture dependencies amongst related stocks for prediction [102, 103]

## 6 Conclusions

We applied novel methods in Bayesian neural networks for multi-step-ahead stock price forecasting before and during first phase of COVID-19. The Bayesian neural network used state-of-art sampling strategy that incorporated parallel computing, Langevin gradients and parallel tempering MCMC for improving sampling which provided very promising results when compared to novel neural networks methods. Our investigation revealed that it is important to incorporate data during an extreme event for better model building. In the experiments, the data from initial phase of COVID-19 in the training dataset improved the prediction accuracy is significantly. Hence, high volatility in the stock blackprice makes forecasting very challenging and increases model uncertainty. Although machine learning methods provide accurate prediction, their applicability in stock price prediction remains due to volatility and hence model validity is important. With robust uncertainty quantification via Bayesian inference, investors would find more confidence in predictions using Bayesian neural networks.

The methodology provided better prediction performance prior to the COVID-19 pandemic which is not surprising given the market crash. The results show that Bayesian neural networks provide reasonable predictions with robust uncertainty quantification despite high market volatility during the first phase of the COVID-19 pandemic. This paper provides motivation for multivariate forecasting approaching using Bayesian deep learning methods which could improve the results further.

## 7 Data and software

We provide open-source Python code and data for further experiments and extensions: https://github.com/sydney-machine-learning/Bayesianneuralnet_stockmarket.

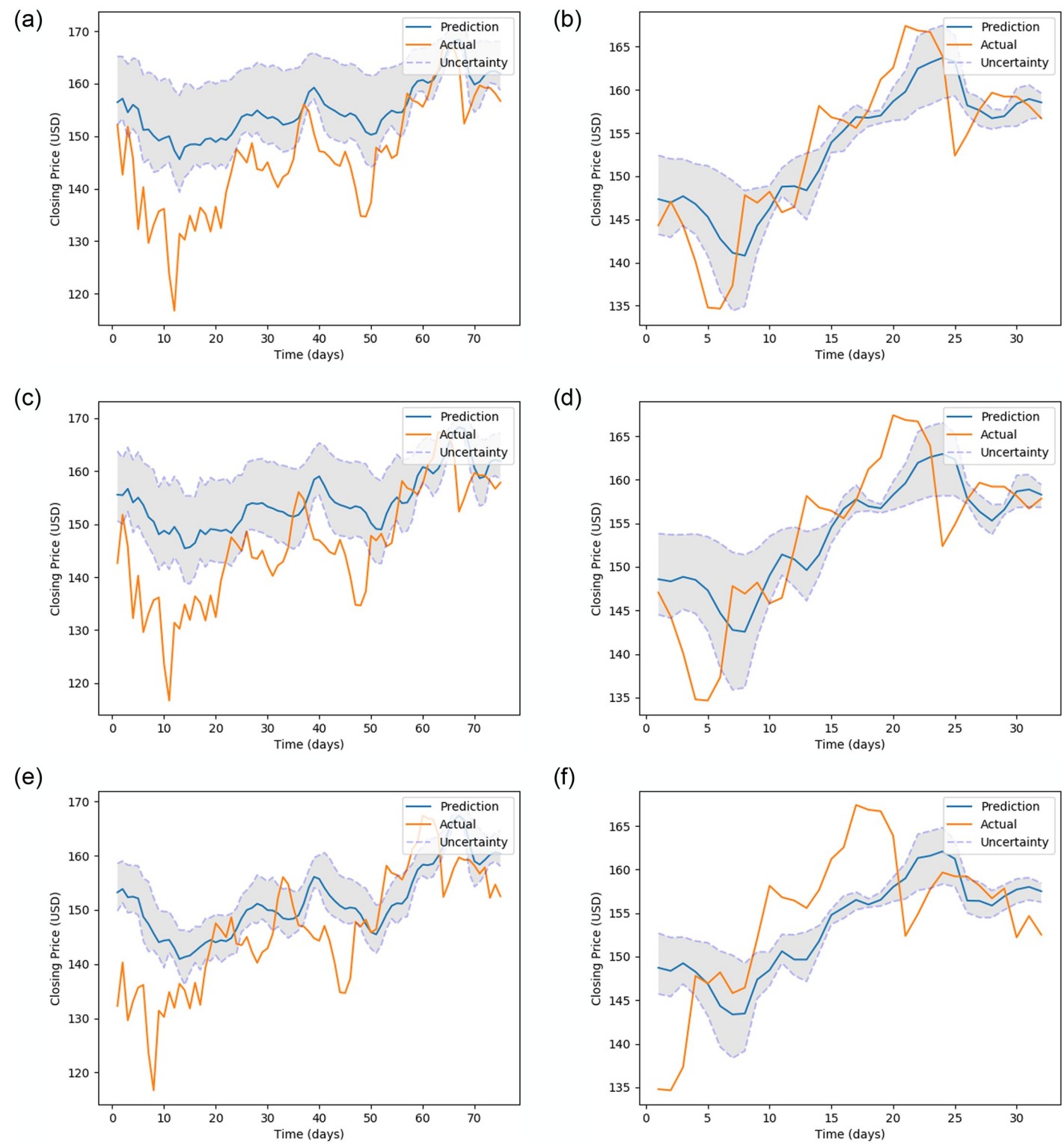

**Fig 10. Prediction and uncertainty over test data of MMM (data setup 1 vs data setup 2).**

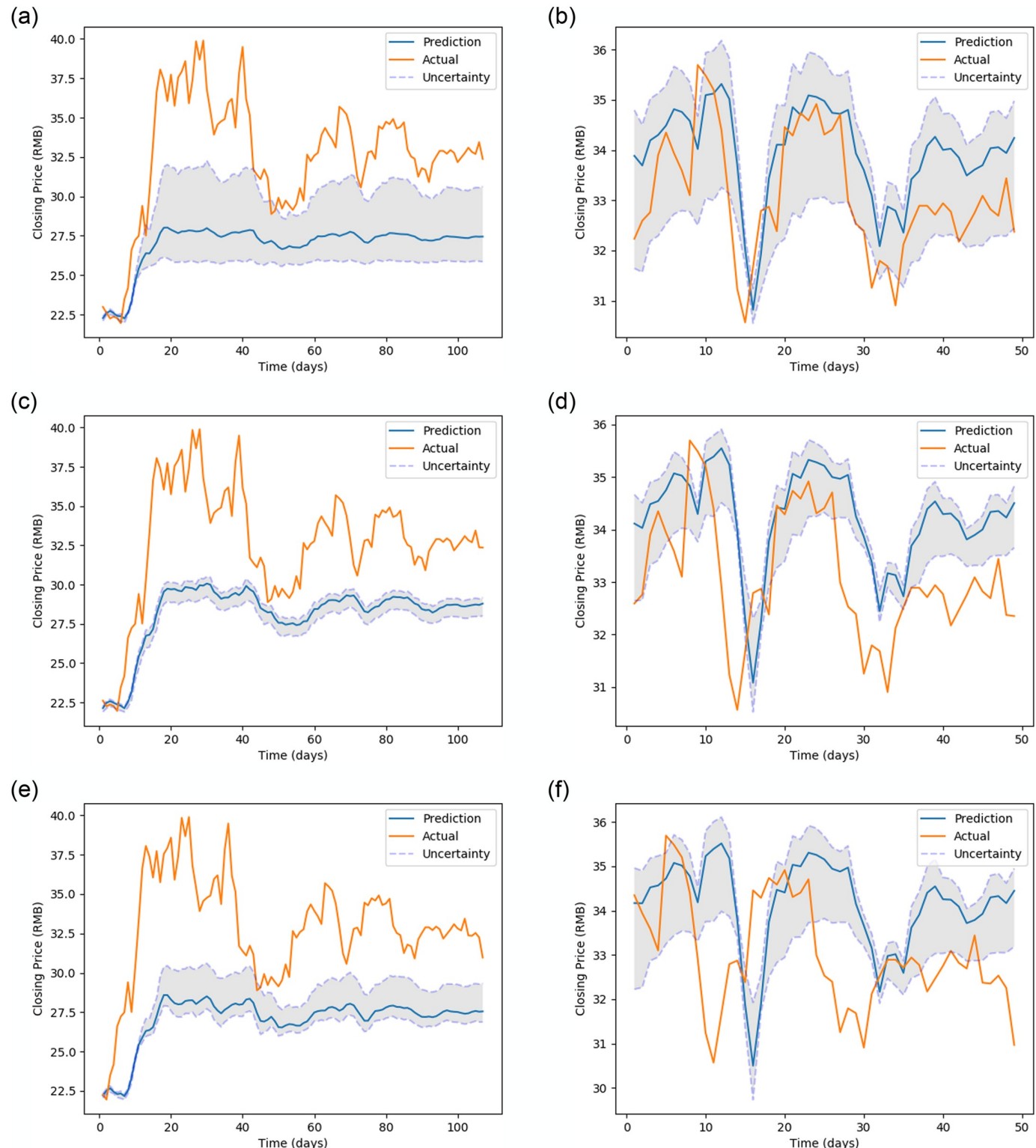

**Fig 11. Prediction and uncertainty over test data of 600118.SS (data setup 1 vs data setup 2).**

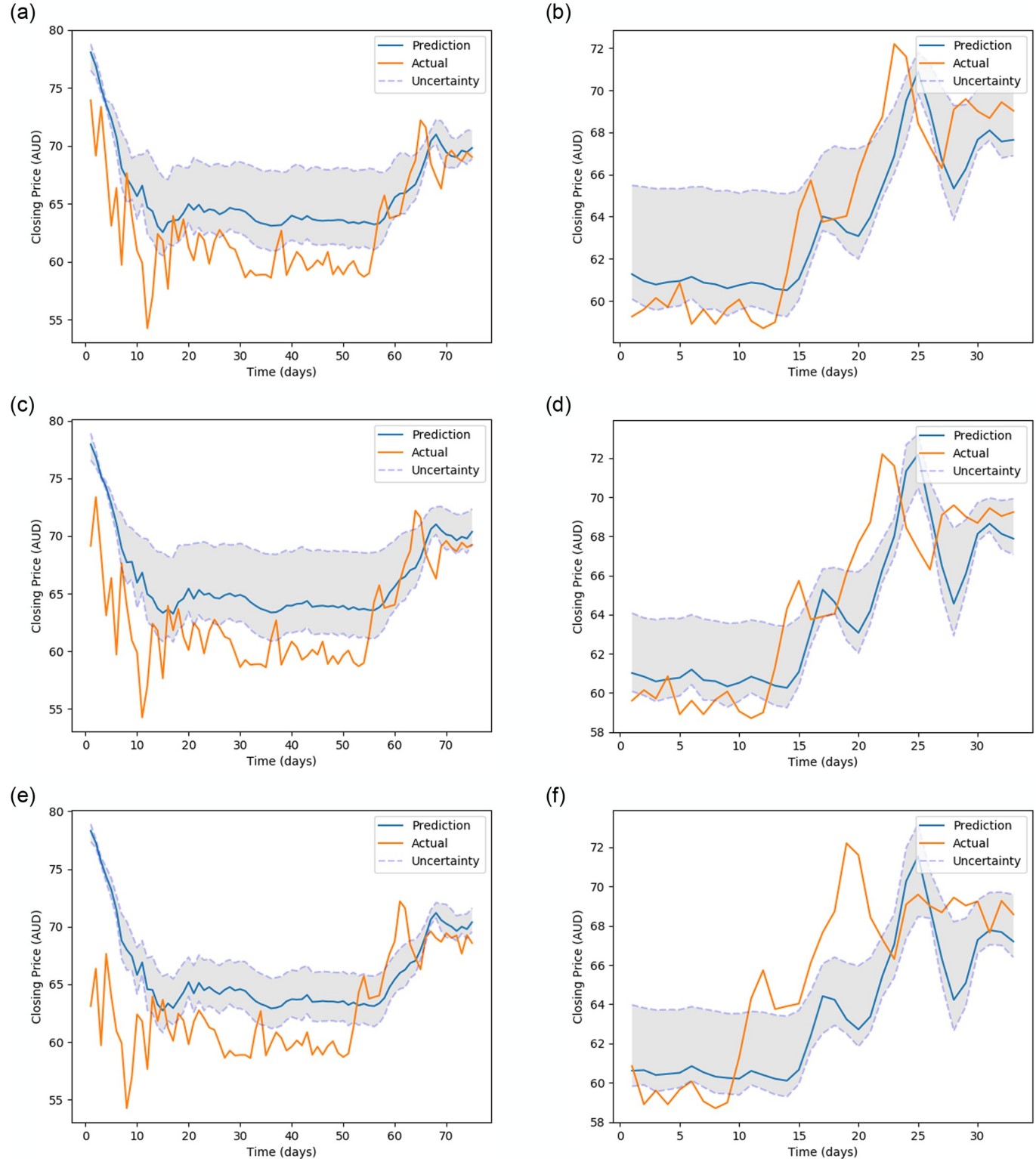

**Fig 12. Prediction and uncertainty over test data of CBA.AX (data setup 1 vs data setup 2).**

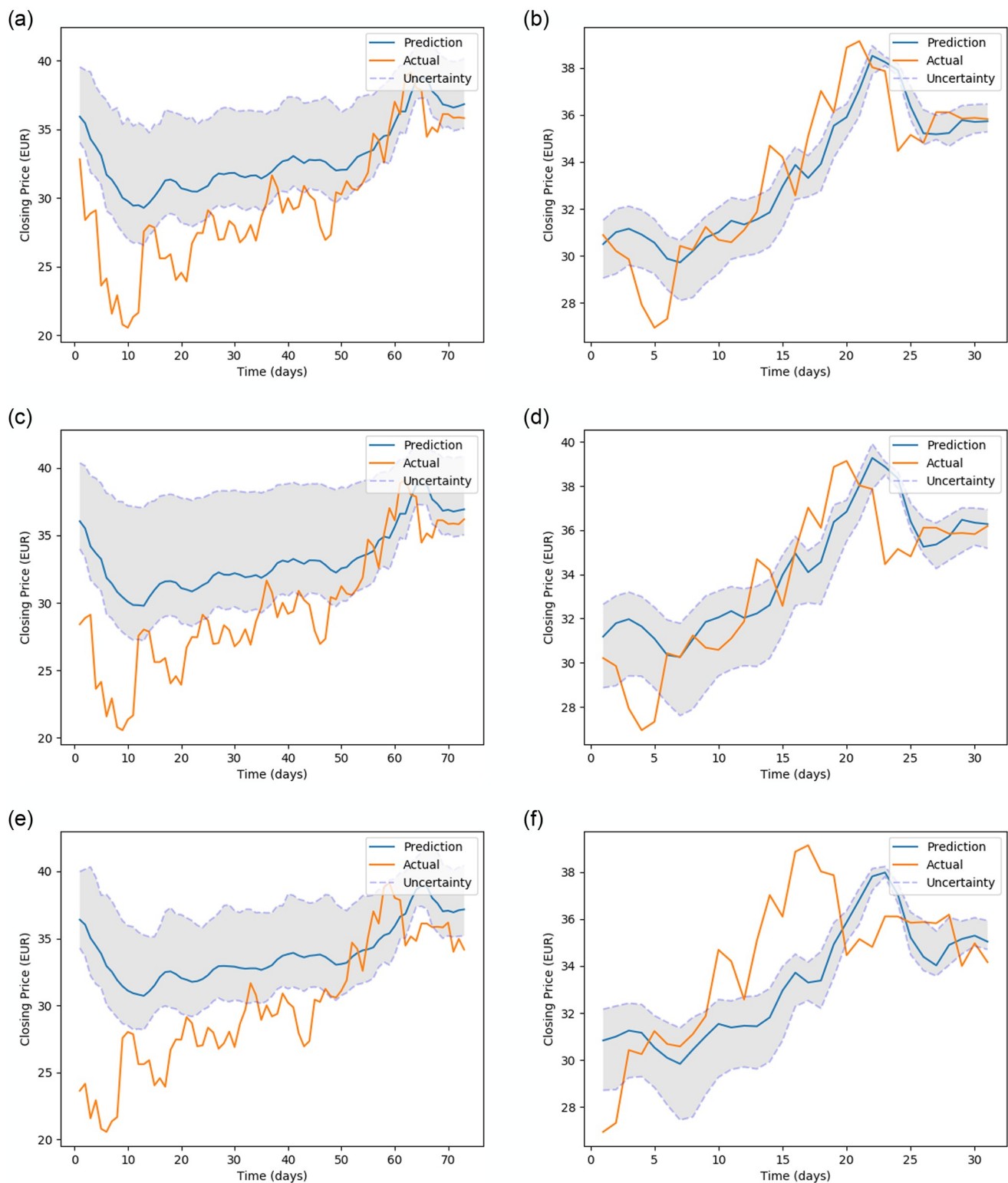

**Fig 13. Prediction and uncertainty over test data of DAI.DE (data setup 1 vs data setup 2).**

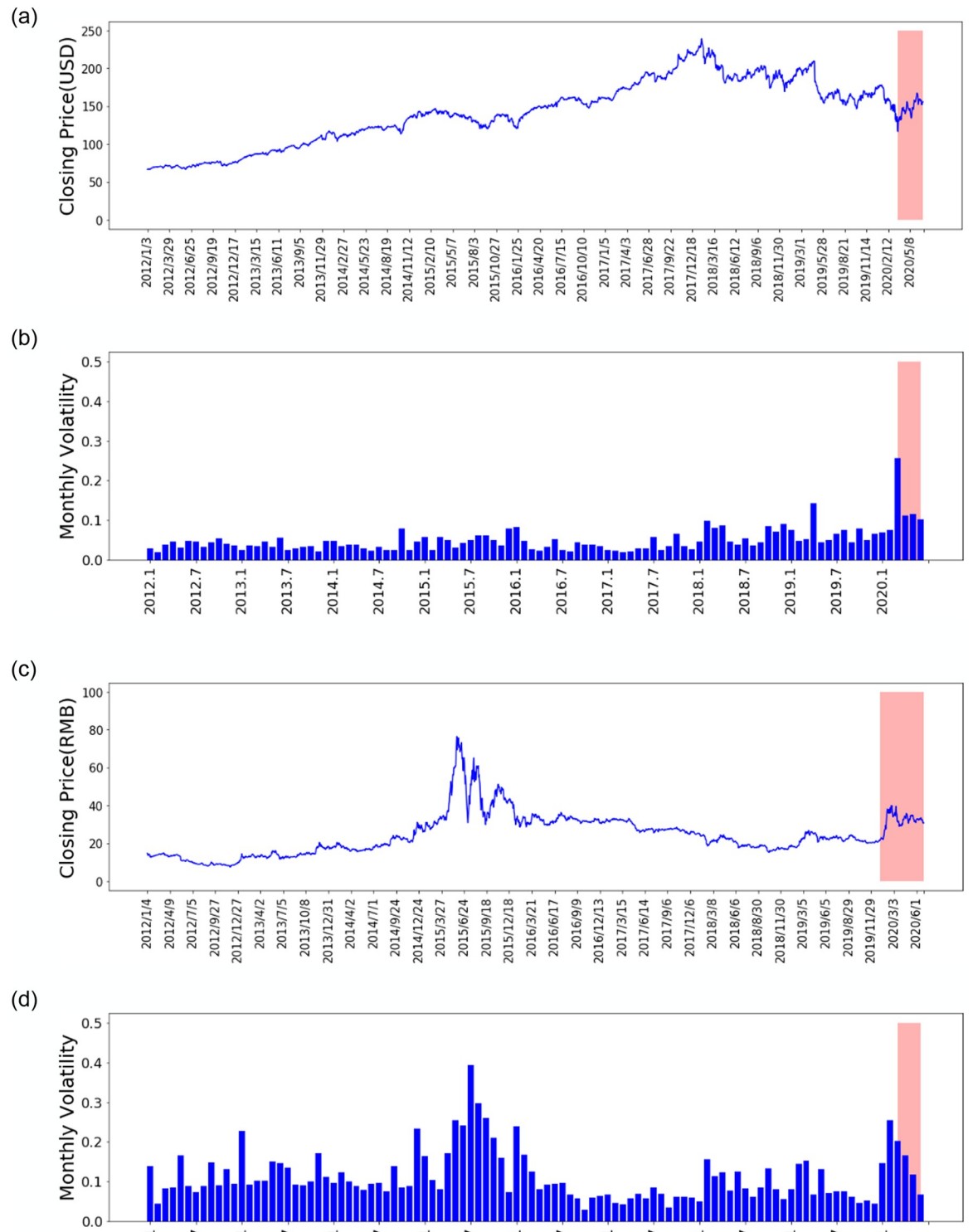

**Fig 14. Stock price time series and monthly volatility for stocks MMM and 600118.SS.**

(a)

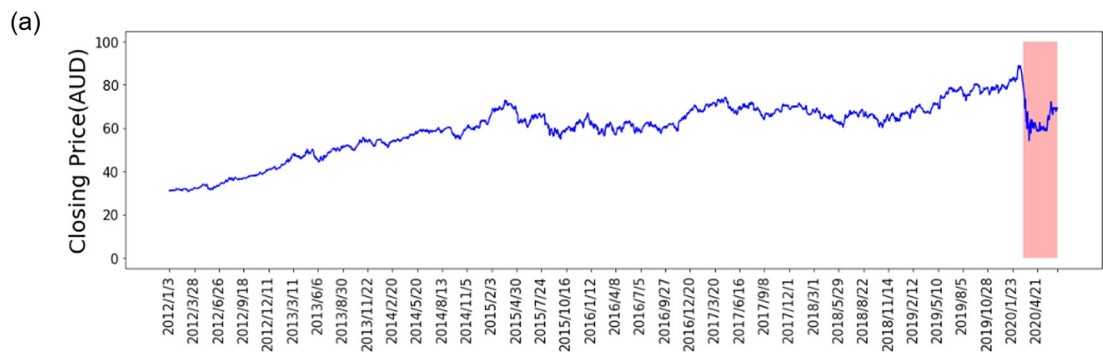

(b)

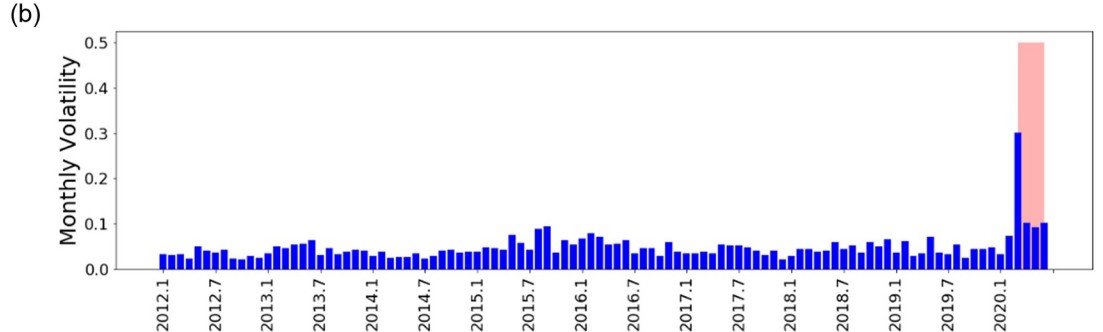

(c)

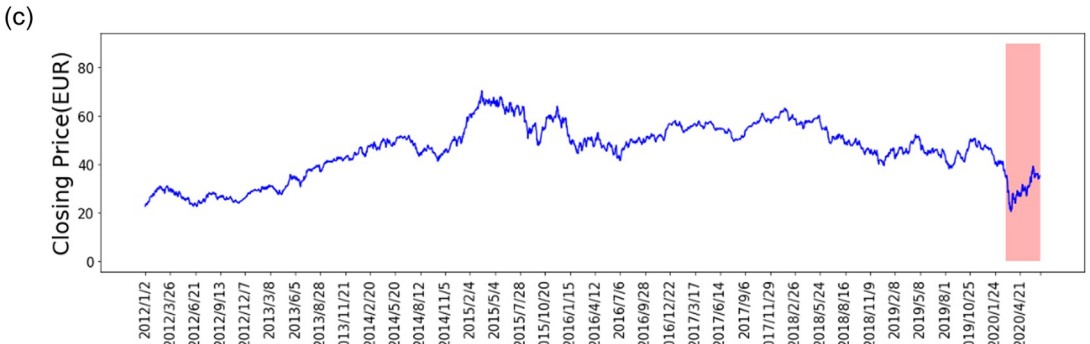

(d)

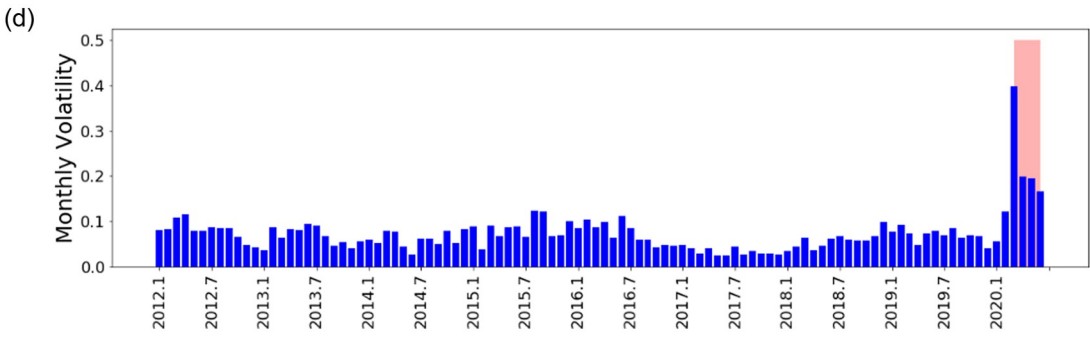

**Fig 15. Stock price time series and monthly volatility for stock CBA.AX and DAI.DE.**

## Author Contributions

**Conceptualization:** Rohitash Chandra.

**Methodology:** Rohitash Chandra, Yixuan He.

**Supervision:** Rohitash Chandra.

**Visualization:** Yixuan He.

**Writing – review & editing:** Rohitash Chandra.

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
