## [Decision Letter · Decision Letter 0]

30 Apr 2021

PONE-D-21-10570

Bayesian neural networks for stock market forecasting  before and during COVID-19 pandemic

PLOS ONE

Dear Dr. Chandra,

Thank you for submitting your manuscript to PLOS ONE. After careful consideration, we feel that it has merit but does not fully meet PLOS ONE’s publication criteria as it currently stands. Therefore, we invite you to submit a revised version of the manuscript that addresses the points raised during the review process.

We look forward to receiving your revised manuscript.

Kind regards,

Junhuan Zhang, PhD

Academic Editor

PLOS ONE

Journal Requirements:

2. Please ensure that you refer to Figure 5-8 and 10-13 in your text as, if accepted, production will need this reference to link the reader to the figure.

Reviewers' comments:

Reviewer's Responses to Questions

**Comments to the Author**

1. Is the manuscript technically sound, and do the data support the conclusions?

Reviewer #1: Partly

Reviewer #2: Yes

2. Has the statistical analysis been performed appropriately and rigorously? 

Reviewer #1: Yes

Reviewer #2: No

3. Have the authors made all data underlying the findings in their manuscript fully available?

Reviewer #1: Yes

Reviewer #2: Yes

4. Is the manuscript presented in an intelligible fashion and written in standard English?

Reviewer #1: No

Reviewer #2: Yes

5. Review Comments to the Author

Reviewer #1: The abstract should be reformulated. This is a very important part of the article. The authors spend too much writing on background. Instead, the authors should better present their research, the main contributions and results, and the conclusions that might be drawn from these results.

In my opinion, the introduction is not well focused. Background should be brief. The originality (novelty) and relevancy of the study should be established with better efforts. My understanding is that this study is an empirical one. Thus, it is expected that the introduction should also include hypothesis and objectives of the study, followed by a justification of the methodology. The current manuscript needs much improvements on all these aspects.

In the introduction, authors claim to study stock market prediction using Bayesian neural networks. However, from the data description, I see the study actually only considers 4 stocks, each from a different country though. Predicting the stock price of a particular company is a very different task from stock market prediction. I believe the authors need more careful wording, and should articulate their research question in the introduction.

More on the data, I don't see any explanation as to why these companies are selected for experiments. Are there any criteria? Also, how to decide the date after which the stock price is affected by COVID-19. Moreover, the authors choose different dates for different stocks/countries. Reasons and justifications are expected here.

The abbreviation FNN-Adam and FNN-SGD are used without mentioning full names. Please check if the use of other abbreviations is of the same problem.

I think one of the main tasks of this study is to compare Bayesian neural network with other neural network methods in terms of forecasting performance. Is it sound to consider only one performance measure (RMSE) here?

From my perspective, the conclusion is quite weak. A more detailed conclusion is needed. The novel method applied here does not seem to outperform state-of-art machine learning methods, at least I can't see it from the conclusion part. The better performance prior COVID-19 is no surprise indeed, and thus does not add any weight to the conclusion part. I expect to see more intelligent conclusions such as the real advantages of this novel Bayesian neural network method. Probably, a comparison with other state-of-art studies would be helpful. This can also be added to the discussion part.

English writing must be carefully revised. Usually, use of WE/OUR in the academic writing should be avoided. I have encountered many grammar mistakes and typos while reading. I list them as below but there are probably more in the manuscript. Thought I am not a native speaker, I feel like the manuscript would benefit from a proof read by a native speaker.

Mistakes I have spotted:

Line 16, “Markov Chain Monte Carlo (MCMC) methods provides a means…”

Line 17, “As the size of model and data continues increases…”

Line 205-206, “The probabilistic neural network model employs the posterior distribution to provides uncertainty quantification on the predictions.”

Line 293, “…we set the burn-in rate is 0.5”

Line 376, “We good prediction accuracy is needed not just for the day ahead, …”

Line 430, “COVID-19 which is not surprising given internal market-crush”

Line 431, “…its is more challenging to provide forecasting during COVID-19…”

Reviewer #2: This paper applies a Bayesian neural networks for multi-step-ahead stock market forecasting before and during COVID-19. But in this version，I don't think the author has made it clear where their novelty ies, is it the novelty of method, or is it the novelty of predicting the stock price changes before and after covid-19? In the paper, the author mentioned that “In the literature, there has not much been done using Bayesian neural networks for stock markets that features robust uncertainty quantification. We can use them to harness power of neural networks that provides good prediction accuracy and also quantify uncertainty. Moreover, there has not been much work that shows how robust machine learning models such as neural networks perform post COVID-19 given major changes in the international stock market with disruptions in international trade and prediction.”，but actually in the listed or not listed references， there are some papers for forecasting in COVID-19. The authors should state exactly the difference of this paper from others, not in a general way. Besides, I list other questions for revision reference.

1. Why Bayes neural network is suitable（even superior）for stock price forecasting. The highlights of this paper should be further addressed.

2. The authors choose 4 stock prices from 4 countries. But I think the selected stocks are not the most representative market stock. Why these datasets are selected should be further clarified.

3. Two other methods are compared with the Bayes neural network, that is, FNN-Adam and FNN-SGD, whose full names should be given when they first appear.

4. The abstract is poorly written. The authors do not clearly show the highlights and significance of this work.

5. The parameter settings in the experiment is very important. I think the authors should give an illustration of parameters in different forecasting models.

6. “”In Setup-2, we include parts of the data during COVID-19 in the training set with all the training data from Setup-1.” What does ”parts” mean here？ I suggest the authors to give an exact time period and data length to show them. Besides， what is the objective of Setup 2.

6. PLOS authors have the option to publish the peer review history of their article (what does this mean?). If published, this will include your full peer review and any attached files.

Reviewer #1: No

Reviewer #2: No

---

## [Author Response · Author response to Decision Letter 0]

6 May 2021

The authors thank reviewers for valuable comments. Pls find the response to review comments attached with the manuscript.

---

## [Decision Letter · Decision Letter 1]

25 May 2021

PONE-D-21-10570R1

Bayesian neural networks for stock market forecasting  before and during COVID-19 pandemic

PLOS ONE

Dear Dr. Chandra,

Thank you for submitting your manuscript to PLOS ONE. After careful consideration, we feel that it has merit but does not fully meet PLOS ONE’s publication criteria as it currently stands. Therefore, we invite you to submit a revised version of the manuscript that addresses the points raised during the review process.

We look forward to receiving your revised manuscript.

Kind regards,

Junhuan Zhang, PhD

Academic Editor

PLOS ONE

Journal Requirements:

Reviewers' comments:

Reviewer's Responses to Questions

**Comments to the Author**

1. If the authors have adequately addressed your comments raised in a previous round of review and you feel that this manuscript is now acceptable for publication, you may indicate that here to bypass the “Comments to the Author” section, enter your conflict of interest statement in the “Confidential to Editor” section, and submit your "Accept" recommendation.

Reviewer #1: All comments have been addressed

Reviewer #2: All comments have been addressed

2. Is the manuscript technically sound, and do the data support the conclusions?

Reviewer #1: Yes

Reviewer #2: Yes

3. Has the statistical analysis been performed appropriately and rigorously? 

Reviewer #1: Yes

Reviewer #2: N/A

4. Have the authors made all data underlying the findings in their manuscript fully available?

Reviewer #1: Yes

Reviewer #2: Yes

5. Is the manuscript presented in an intelligible fashion and written in standard English?

Reviewer #1: Yes

Reviewer #2: Yes

6. Review Comments to the Author

Reviewer #1: The revised manuscript has addressed all my comments. I only have one additional suggestion. In the study, the authors use the novel method to predict stock prices of 4 individual companies, rather than market indices. I think it is more appropriate to describe it as "stock price forecasting" than "stock market forecasting". Thus, I suggest to revise the title and relevant parts in the main text.

Reviewer #2: The authors have carefully revised paper accorrdingt to my comments and other reviewr's comment. I suggest an

acceptance.

7. PLOS authors have the option to publish the peer review history of their article (what does this mean?). If published, this will include your full peer review and any attached files.

Reviewer #1: No

Reviewer #2: No

---

## [Author Response · Author response to Decision Letter 1]

27 May 2021

We attached it with the manuscript.

---

## [Editor Report · Decision Letter 2]

31 May 2021

Bayesian neural networks for stock price forecasting  before and during COVID-19 pandemic

PONE-D-21-10570R2

Dear Dr. Chandra,

We’re pleased to inform you that your manuscript has been judged scientifically suitable for publication and will be formally accepted for publication once it meets all outstanding technical requirements.

Kind regards,

Junhuan Zhang, PhD

Academic Editor

PLOS ONE

---

## [Editor Report · Acceptance letter]

17 Jun 2021

PONE-D-21-10570R2 

Bayesian neural networks for stock price forecasting before and during COVID-19 pandemic 

Dear Dr. Chandra:

I'm pleased to inform you that your manuscript has been deemed suitable for publication in PLOS ONE. Congratulations! Your manuscript is now with our production department. 

Kind regards, 

on behalf of

Dr. Junhuan Zhang 

Academic Editor

PLOS ONE